# EEG decoders track memory dynamics

Yuxuan Li[1,2], Jesse K. Pazdera ◉[1,3] & Michael J. Kahana[1] ✉

Encoding- and retrieval-related neural activity jointly determine mnemonic success. We ask whether electroencephalographic activity can reliably predict encoding and retrieval success on individual trials. Each of 98 participants performed a delayed recall task on 576 lists across 24 experimental sessions. Logistic regression classifiers trained on spectral features measured immediately preceding spoken recall of individual words successfully predict whether or not those words belonged to the target list. Classifiers trained on features measured during word encoding also reliably predict whether those words will be subsequently recalled and further predict the temporal and semantic organization of the recalled items. These findings link neural variability predictive of successful memory with item-to-context binding, a key cognitive process thought to underlie episodic memory function.

Cognitive neuroscience has experienced a paradigm shift. After early studies appeared to localize cognitive operations to specific brain regions, subsequent work challenged this view, favoring distributed models of cognitive processes. Multivariate statistical methods, including classification and similarity-based approaches, now enable researchers to test models of these distributed processes and their dynamics. Here we apply these methods to episodic memory, with the goal of testing predictions of cognitive theories regarding the dynamics of encoding- and retrieval-related cognitive processes.

Our approach uses multivariate analysis of scalp electro-encephalographic (EEG) recordings to classify single-trial accuracy of memory encoding and retrieval. Prior research has demonstrated that classifiers trained on spectral features extracted from scalp EEG recordings can predict recognition memory performance[1,2]. Building on this work, we first asked whether such classifiers can distinguish the encoding of subsequently recalled and non-recalled items in a verbal free recall task. Applying the same classification approach to recall period data, we further asked whether, prior to vocal free recall, classifiers could distinguish correct recalls from intrusion errors. Obtaining an affirmative answer to these questions allowed us to assess whether the encoding and retrieval classifiers' low-dimensional neural representation could illuminate the dynamics of memory processes. Specifically, can encoding and retrieval classifiers differentiate items recalled on the basis of semantic and/or temporal associations, and do the classifiers exhibit dynamics consistent with the predictions of memory theory? To the extent that encoding success relies upon associating items with their temporal and/or semantic context[3–5],

we predicted that the encoding classifier would assign greater probabilities to spectral features of EEG signals recorded during the encoding of subsequently clustered items. To the extent that cognitive resources deplete over the course of the encoding and retrieval periods[6,7], we predicted that classifier output probabilities would decline during these intervals.

To probe the upper-bound of what could be learned using scalp EEG methods, we assembled a dataset comprising more than one million word encoding events and nearly a half-million delayed recall responses. Ninety-eight healthy young adults each came to the lab for 24 days of testing. The large number of trials and sessions contributed by each participant allowed us to train classifiers at the individual trial level and validate these classifiers using data from held-out sessions (rather than held-out lists).

Prior intracranial (iEEG) studies have documented increases in high-frequency activity (HFA) across a network of cortical and subcortical regions predicting successful memory encoding and retrieval[8–10] including the hippocampus[11]. These studies found that decreases in low-frequency activity, particularly in the 8–13 Hz alpha band, accompanied the HFA increases. Whereas the previously cited studies examined the univariate relation between spectral EEG features and memory processes, other studies have taken a multivariate approach, using machine learning classifiers to distinguish single-trial encoding of subsequently remembered and forgotten items. These multivariate studies have found a similar reliance on combined HFA and LFA effects in predicting memory success[12–16]; two of these studies[13,16] also showed that multivariate classifiers could distinguish

[1]Department of Psychology, University of Pennsylvania, Philadelphia, PA, USA. [2]Present address: Department of Psychology, Stanford University, Stanford, CA, USA. [3]Present address: Department of Psychology, Neuroscience & Behaviour, McMaster University, Hamilton, ON, Canada. ✉e-mail: kahana@psych.upenn.edu

moments preceding correct recalls from periods of deliberation that did not lead to successful recall. Non-invasive EEG and MEG studies have also found that greater HFA and diminished alpha-band activity generally accompany successful as compared with unsuccessful memory encoding and retrieval[17–21]; theta-band activity exhibits increases in some studies and decreases in other studies[17–23]. A review of the conflicting theta effects[24] suggests that increases may appear in contrasts that look specifically at lower theta frequencies, isolate specific associative retrieval processes, and/or separate broadband from narrowband EEG components[11,25].

iEEG classifiers rely on differences in varied spectral features to predict memory success. However, two key limitations in previous work have made it difficult to distinguish the contributions of these spectral features. First, comparisons between subsequently remembered and non-remembered encoding events often pooled events across all list serial positions. Because people typically exhibit better memory for items appearing in early and late serial positions, any signal that differs between the encoding periods of these items could predict mnemonic success when comparisons pool across serial positions. Results obtained using such comparisons cannot distinguish neural signals that predict encoding success of an item learned in a given position with those that only predict differences across positions in the list, independent of encoding success. A similar problem exists across retrieval output positions, as people make more recalls early during a free recall period and comparisons that pool across the entire recall period will not be able to distinguish correlates of retrieval success at a given output time with those that simply distinguish early vs. late times within an extended retrieval period.

A second limitation concerns the memory retrieval signals just prior to spoken free recall. Comparisons between accurate recall and deliberation intervals do not control for the premotor activity related to word vocalization, even though these two types of retrieval events were carefully matched in their temporal distributions over the retrieval period. This could be especially important when generalizing findings from intracranial recordings to non-invasive measures. Building on iEEG studies that examined HFA increase as associated with retrieval success by comparing correct and incorrect recall responses[10], here we have the opportunity to examine whether increased HFA is similarly seen in trial-level retrieval classification with scalp EEG, in a dataset with rich behavioral responses that affords the possibility of revealing individual differences beyond group-level consistency.

Targeting these two limitations, we highlight a novel approach to training episodic memory success classifiers during encoding and retrieval. To disentangle signals predictive of memory success from signals associated with early versus late encoding or retrieval, we re-sampled encoding and retrieval events when training the classifiers to ensure a matching ratio of positive and negative examples from each serial position or recall output position. To eliminate any confound with pre-motor activity in predicting retrieval success, we tested whether classifiers could distinguish, prior to vocal free recall, whether a remembered item appeared on the study list (i.e., distinguishing correct recalls from intrusion errors).

We contrasted classifiers trained on re-sampled events with classifiers trained on all events (as in previous studies) and asked if the two complementary training approaches encouraged the models to learn different signals predictive of mnemonic states, as related to the cognitive theories discussed above. We expected that training the encoding classifiers with re-sampled events would induce a greater focus on differentiating signals underlying item encoding irrespective of serial positions. Consequently, these classifiers would assign greater output probabilities to spectral features associated with the encoding of subsequently temporally- or semantically-clustered items. In contrast, classifiers trained with all events would learn a combination of clustering-based and primacy-based signals, and they would therefore

show a decline in classifier output over list positions. Each of these two hypotheses has a counterpart during the retrieval phase, which can be tested using classifiers trained to distinguish correct recalls from recall intrusions or errors.

## Results

We first sought to characterize the spectral correlates of successful memory encoding and retrieval in delayed free-recall. Comparing spectral power estimates during the 1.6 s encoding epochs of subsequently recalled and non-recalled items (the subsequent memory effect, or SME) identified spectral markers of successful encoding. Similarly, comparisons of power in the 500 ms epochs preceding correct recalls and recall errors (prior-list and extra-list intrusions) identified spectral markers of successful contextual retrieval.

### Spectral correlates of encoding success

Tasks such as free recall exhibit pronounced differences in the recallability of items studied in early, middle, and late serial positions (the well-known primacy and recency effects, illustrated in Fig. 1a for our experiment). As such, subsequently recalled items will tend to come from favorable list positions, introducing a potential confound into the traditional subsequent memory contrast which simply compares neural activity during the encoding of subsequently remembered and non-remembered items. To the extent that neural activity exhibits distinct relations to list position and to encoding success, subsequent memory comparisons will conflate these effects. To address this potential confound, we introduce an event re-sampling procedure that equalizes the contribution of recalled and non-recalled items across list positions. This method allows us to isolate EEG correlates of mnemonic variability at a given list position from EEG correlates of serial position that may reflect a mixture of mnemonic and non-mnemonic processes (see Methods).

Using event re-sampling, we compared the distributions of spectral power estimates during the encoding epochs of subsequently recalled and non-recalled items. This comparison generated a $t$-statistic for each participant, which served as a normalized estimate of that participant's effect size. We then conducted group-level statistics on the distribution of individual participant $t$-statistics. These comparisons revealed that group-level decreases in 10−30 Hz activity marked encoding success (Fig. 1d, left). In anterior-superior regions of interest (ROIs), memory encoding-related power decreases extended into higher frequencies, and at posterior electrodes decreases extended into lower frequencies.

Analyzing all encoding events without re-sampling yielded the more typical EEG signature of successful encoding: Widespread increases in HFA, decreases in LFA, and a positive theta SME in anterior regions collectively marked periods of encoding success (Fig. 1e, left). The differences in spectral signatures of encoding success across these two methods echoed prior work showing that HFA decreases and theta increases from early to late serial positions[21,26].

### Spectral correlates of retrieval success

We applied a similar event re-sampling method in our analysis of retrieval success, as seen in the difference between correct recalls and intrusions during the 500 ms immediately preceding vocalization. Due to EEG fluctuations over the course of the recall period, and because errors tend to occur more at later output times, event re-sampling helps isolate the neural correlates of retrieval success while equating output time/position. We found that increases in HFA reliably distinguished intrusions and correct recalls both when recall responses were re-sampled to match recall accuracy rates across the 7.5 s recall bins and when compared across all correct recall responses and recall errors (Fig. 1d, e, right panels). That scalp-recorded HFA signals distinguished true from false recall suggests that non-invasive measurements of HFA can pick up subtle variation in the neuronal correlates of

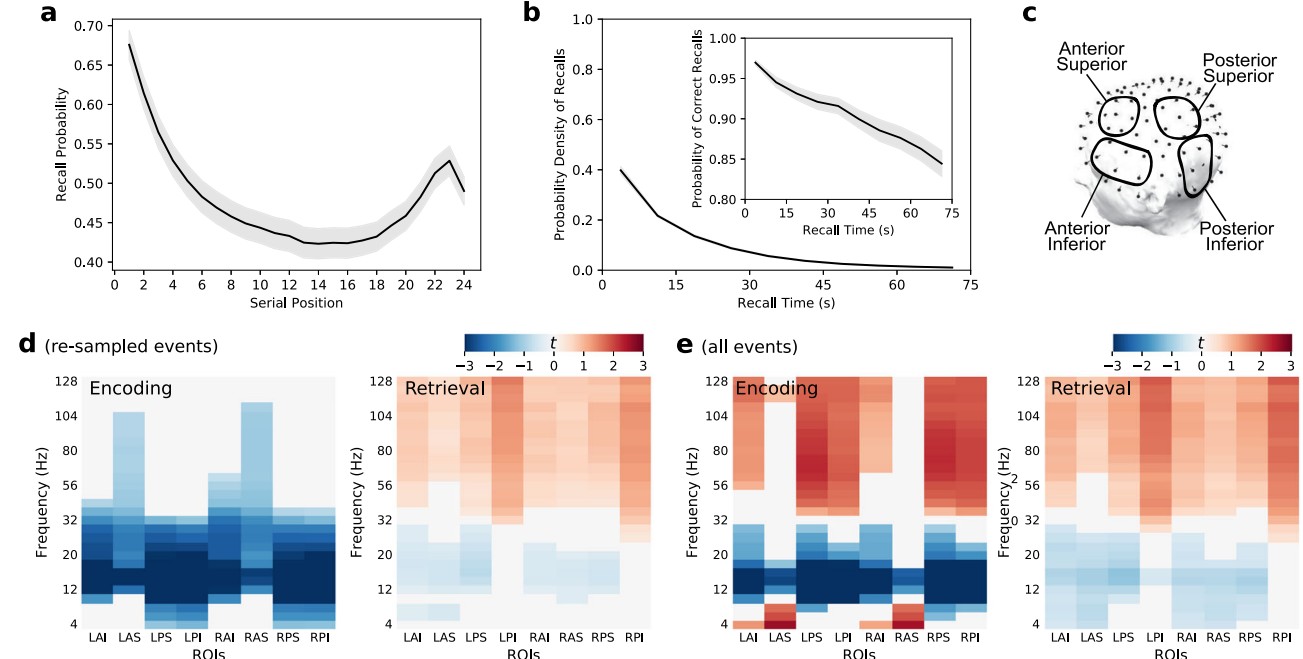

**Fig. 1 | Behavioral responses and spectral markers of successful encoding and retrieval. a** Probability of delayed recall exhibited a strong primacy effect and a small recency effect, replicating classic studies. The arithmetic distractor task performed between the final study item and the recall period attenuated the recency effect which normally exceeds the primacy effect in immediate free recall[29]. **b** The probability density of recalls (binned every 7.5 s) declined throughout the retrieval period, replicating classic studies. The subpanel illustrates the proportion of responses that were correct. **c** Regions of interest. **d** Differences in spectral power during encoding of subsequently remembered words versus subsequently forgotten words (left), and for 500 ms intervals preceding correct recalls versus intrusions (right). Contrast computed with event re-sampling to ensure matching ratio of successful and unsuccessful memory events from each serial position or recall bin. Colors in each frequency × ROI pair correspond to the mean participant-specific independent *t*-statistics. Significant group-level increases and decreases (false discovery rate corrected) are shown in red and in blue, respectively. **e** Visualization as in (**d**), but across all successful and unsuccessful memory events. L: left, R: right, A: anterior, P: posterior, I: inferior, S: superior. Error shades in (**a**, **b**) indicate the standard error of the mean across participants. Source data are provided as a Source Data file.

contextual retrieval. To our knowledge, only studies involving indwelling electrodes succeeded in distinguishing correct recalls from intrusion errors prior to this report[10,27].

### Individual-differences in group-level effects

Although group-level analyses revealed consistent spectral markers of mnemonic encoding and retrieval success, one can see variability in these effects across individuals, especially for the HFA component of the subsequent memory effect under event re-sampling (Fig. 2a). When analyzing all events, however, one sees a greater degree of consistency across participants in all three effects: theta increases, alpha / beta decreases, and HFA increases. Spectral markers of successful retrieval exhibit a higher degree of consistency across participants under both event re-sampling and when considering all retrieval events (Fig. 2b). As we designed this experiment with the goal of modeling memory dynamics at the individual participant level, our next set of analyses trained and cross-validated multivariate, participant-specific decoders of encoding and retrieval success.

### Multivariate classification of encoding and retrieval success

We trained participant-specific multivariate classifiers to decode successful encoding and retrieval epochs based on spectral features across all electrodes. We labeled encoding epochs on the basis of subsequent correct recall and we labeled pre-vocalization retrieval epochs on the basis of list membership, contrasting correct recalls with intrusion errors. We used a leave-one-session-out cross-validation scheme, evaluating classifier performance as the area under the receiver operating characteristic (ROC) curve, or AUC, over the predicted probabilities of events in the held-out sessions (see Methods).

As in the preceding analyses, we report findings both using event re-sampling and with all events.

Before presenting group-level analyses for the full dataset, we first illustrate classifier output for events in example test lists from three participants, as predicted by the corresponding classifiers trained on one of the 20 independent rounds of event re-sampling. As shown in Fig. 3a, the trained classifiers extracted variations in memory states throughout the encoding and retrieval phase. All but one of the encoding and retrieval classifiers for these three participants yielded reliable classification across all held-out test sessions. Comparing the observed AUCs derived from the prediction on all held-out events with the distributions of baseline AUCs obtained from classifiers trained with randomly-shuffled training labels (within-session permutation), five of the six classifiers performed significantly above chance (*p*'s < 0.01; Fig. 3b illustrates results pooled across all 20 runs of event re-sampling).

Figure 4 summarizes classifier performance across all participants. Encoding classifiers reliably predicted performance in held-out sessions both when trained on re-sampled events (mean AUC = 0.59, *SE* = 0.004, Fig. 4a) and when trained on all events (mean AUC = 0.60, *SE* = 0.004, Fig. 4c). In both cases, the encoding classifiers reliably predicted memory success for nearly all of the 88 analyzed participants (87 significant event re-sampling classifiers, 88 significant all-events classifiers, permutation tests). The overall group distribution of observed AUCs far exceeded the group distribution of baseline AUCs from the permutation tests (event re-sampling classifiers: *t*(87) = 24.12, *p* < 0.001; all-events classifiers: *t*(87) = 27.54, *p* < 0.001). Retrieval classifiers also reliably distinguished correct recalls and intrusion errors, both when trained with event re-sampling (mean AUC = 0.57, *SE* = 0.006, 48 significant classifiers) and when trained on all recall events

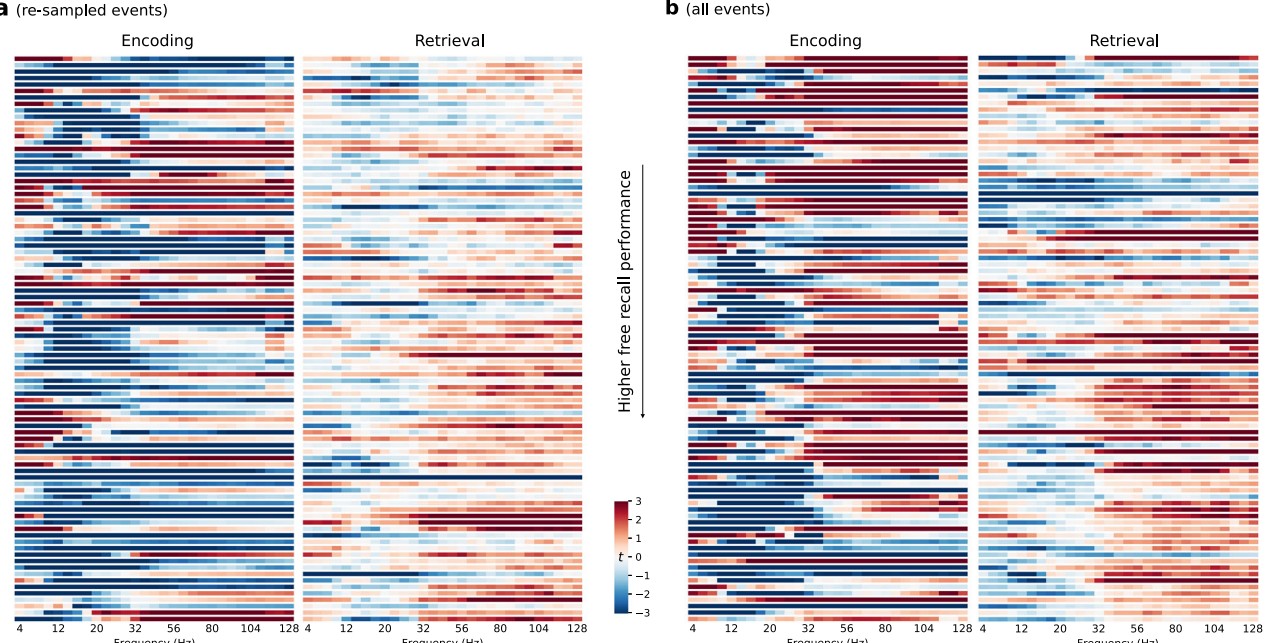

**Fig. 2 | Participant-specific spectral markers of successful episodic memory in encoding and retrieval. a** Individual spectral differences using re-sampled events. **b** Individual spectral differences using all events. Each row shows results from one participant, sorted in ascending recall performance. Participant-specific independent *t*-statistics for the successful and unsuccessful memory comparison are collapsed across eight ROIs. Power increases and decreases are shown in red and in blue, respectively. Source data are provided as a Source Data file.

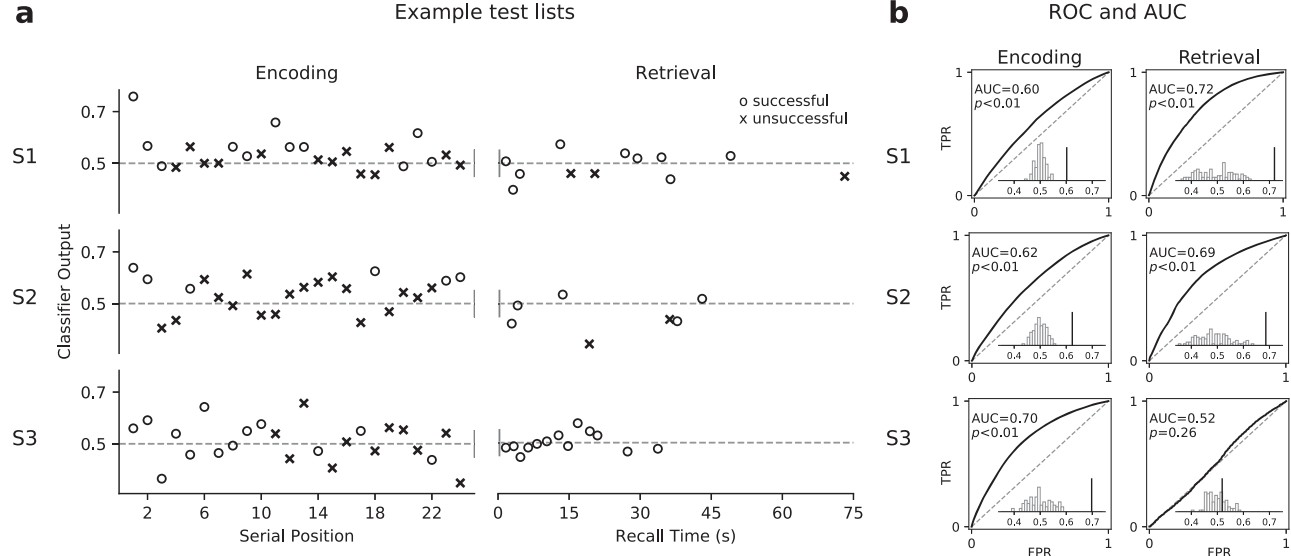

**Fig. 3 | Example classifier prediction and performance. a** Output probabilities (from classifiers trained with event re-sampling) of novel events in example held-out lists as a function of serial position and recall time. Circles mark the ground-truth successful memory instances (subsequently remembered words and correct recalls). X's mark the ground-truth unsuccessful memory instances (subsequently forgotten words and intrusions). **b** ROCs and AUCs across all cross-validation test sessions, averaged over 20 runs of independently re-sampled events. Significance levels were computed by comparing whether the mean observed AUC across runs (marked by the dark line) was greater than a distribution of null AUCs obtained from classifiers trained with permuted training labels (shown in white bars). S1–S3: participant labels. TPR: true positive rate. FPR: false positive rate. Source data are provided as a Source Data file.

(mean AUC = 0.59, *SE* = 0.007, 64 significant classifiers). As with the encoding classifiers, the distribution of retrieval classifier AUCs far exceeded the baseline AUCs obtained from permuted data (event re-sampling classifiers: $t(87) = 10.24$, $p < 0.001$; all-events classifiers: $t(87) = 13.57$, $p < 0.001$).

Classifiers trained with event re-sampling relied on different spectral components in predicting memory success compared to those trained on all events, mirroring our univariate analyses (compare heatmaps in Figs. 1, 4). To estimate the contribution of each frequency × electrode pair to classification performance, we constructed forward models based on the learned weights of the significant individual classifiers[28] and aggregated the activation patterns over electrodes in different ROIs. The significant encoding classifiers consistently relied on decreased LFA when trained to predict single item encoding success in re-sampled events, whereas classifiers trained on all events relied on increased HFA, decreased LFA, and increased

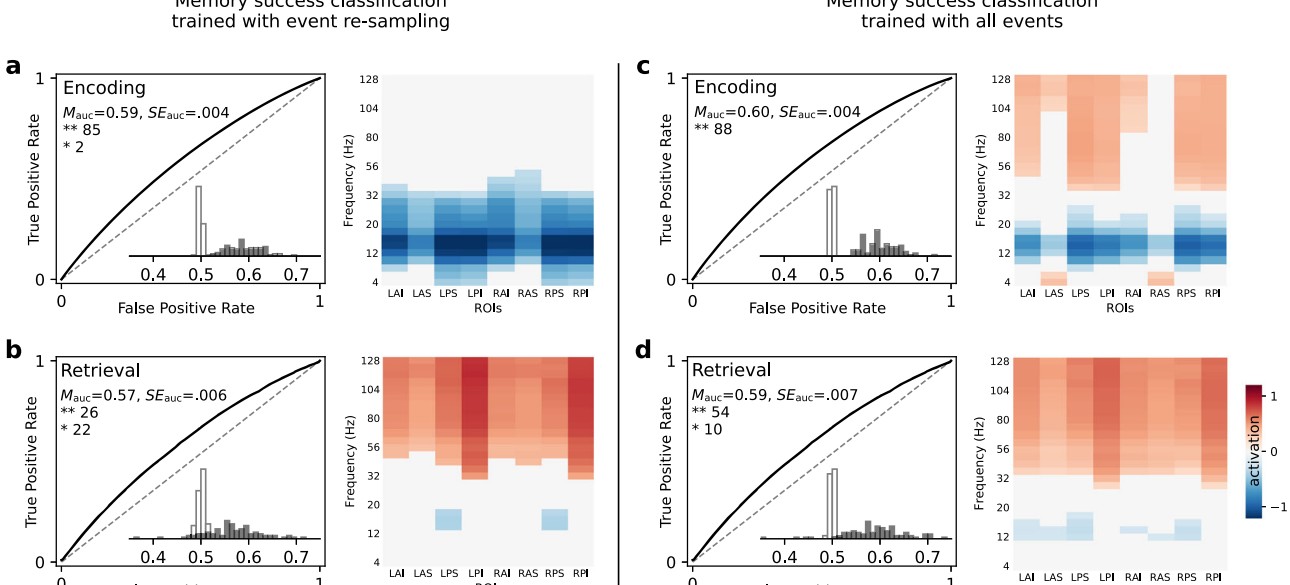

**Fig. 4 | EEG classification of encoding and retrieval success. a** Encoding classifiers trained with event re-sampling predicting subsequently recalled words (see Methods). **b** Retrieval classifiers trained with event re-sampling predicting correct recall responses. **c, d** Encoding and retrieval classifiers trained on all events. **a–d** ROCs depict average results across all participants, with dashed lines indicating chance-level performance. The mean and standard error of the area under the curve performance metric appears in the upper left region along with the number of participants whose classifiers reliably predicted out of sample data. **p < 0.01.

*p < 0.05 (permutation test, as described above in Fig. 3b). Inset histograms indicate the distributions of observed AUCs and baseline AUCs for each participant (see group-level statistical comparisons in main text). Heatmaps to the right of each panel indicate average results of a forward activation model applied to each frequency × ROI pair (see Methods). Significant positive and negative activations (FDR corrected) appear as red and blue, respectively. Source data are provided as a Source Data file.

frontal theta to predict encoding success. Classifiers trained to distinguish correct from incorrect recalls consistently relied on increased HFA and (to a lesser extent) decreased LFA both when trained with re-sampled events and with all events.

Having shown that spectral features can reliably classify imminent recall of intrusions and correct responses, we next examined classification of different intrusion types. Specifically, we asked whether our classification approach could differentiate between prior-list intrusions (PLIs) and extra-list intrusions (ELIs). For this analysis, we trained classifiers using all valid recall responses. Figure 5 shows that classifiers reliably discriminated correct recalls from PLIs (mean AUC = 0.59, SE = 0.007, 57 significant classifiers) and from ELIs (mean AUC = 0.58, SE = 0.008, 50 significant classifiers), with significant classification at the group level (t(87) = 11.86, p < 0.001, and t(87) = 10.13, p < 0.001, respectively, based on group-level t-tests of observed versus baseline AUCs). In both cases, the application of a forward model to the participants with statistically significant classifiers revealed that HFA was greater for correct recalls than for both intrusion types, echoing results shown in Fig. 4b, d, and also that classification of PLIs, but not ELIs, relied on reduced LFA (Fig. 5a, b, right panels). These findings demonstrate that spectral features extracted from scalp EEG recordings can reliably differentiate subtle differences between correct recalls and both PLIs and ELIs.

Although commission of PLIs and ELIs likely involve similar cognitive processes, they differ in their reliance on temporal information (recency) versus semantic similarity[29]. As such, we asked whether spectral features could distinguish these two types of errors. As shown in Fig. 5c, we did not observe reliable group-level classification for this contrast (mean AUC = 0.51, SE = 0.010). But despite poor aggregate level performance (t(87) = 1.63, p = 0.11), data from 14 participants exhibited significant classification. Forward models derived from these 14 significant ELI-PLI classifiers did not yield consistent feature activations for any frequency-ROI pair (see Methods).

## Classifier-based analyses of memory dynamics

Cognitive theories posit that variation in good memory encoding states should predict the degree to which participants exhibit subsequent organization of the learned materials. Specifically, items learned during good encoding states should exhibit strong temporal and semantic clustering[30]. Cognitive studies of memory also indicate that goodness of encoding exhibits a strong primacy gradient over the encoding phase, reflecting rehearsal strategies[31], and/or fatigue of memory encoding networks[6,32].

Encoding classifiers trained on all events will include proportionally more recalled events in early list positions. As such, these classifiers will reflect EEG signals that correlate with list position, whether or not these signals specifically relate to goodness of memory encoding. Classifiers trained with event re-sampling could not leverage such signals and thus provide a purer (though possibly muted) index of encoding success. We hypothesized that classifiers trained with event re-sampling, by isolating cognitive processes underlying encoding success, would predict the subsequent clustering of the recalled items. In contrast, we predicted that classifiers trained with all events would more closely track the primacy gradient, thereby predicting the magnitude of the primacy effect seen in recall data.

To evaluate these hypotheses, we examined whether the encoding classifiers' output probability of the held-out test events tracked memory states across serial positions, as well as this output as a function of subsequent temporal and semantic organization. We therefore sorted encoding events corresponding to subsequently recalled words into four subtypes: (1) BC, both temporally and semantically clustered, (2) TC, temporally clustered, (3) SC, semantically clustered, and (4) NC, not clustered. We designated a studied item as temporally clustered if it was recalled immediately preceding or following the item studied in the preceding serial position; we designated an item as semantically clustered if it was recalled preceding or following a highly semantically similar item (see Methods).

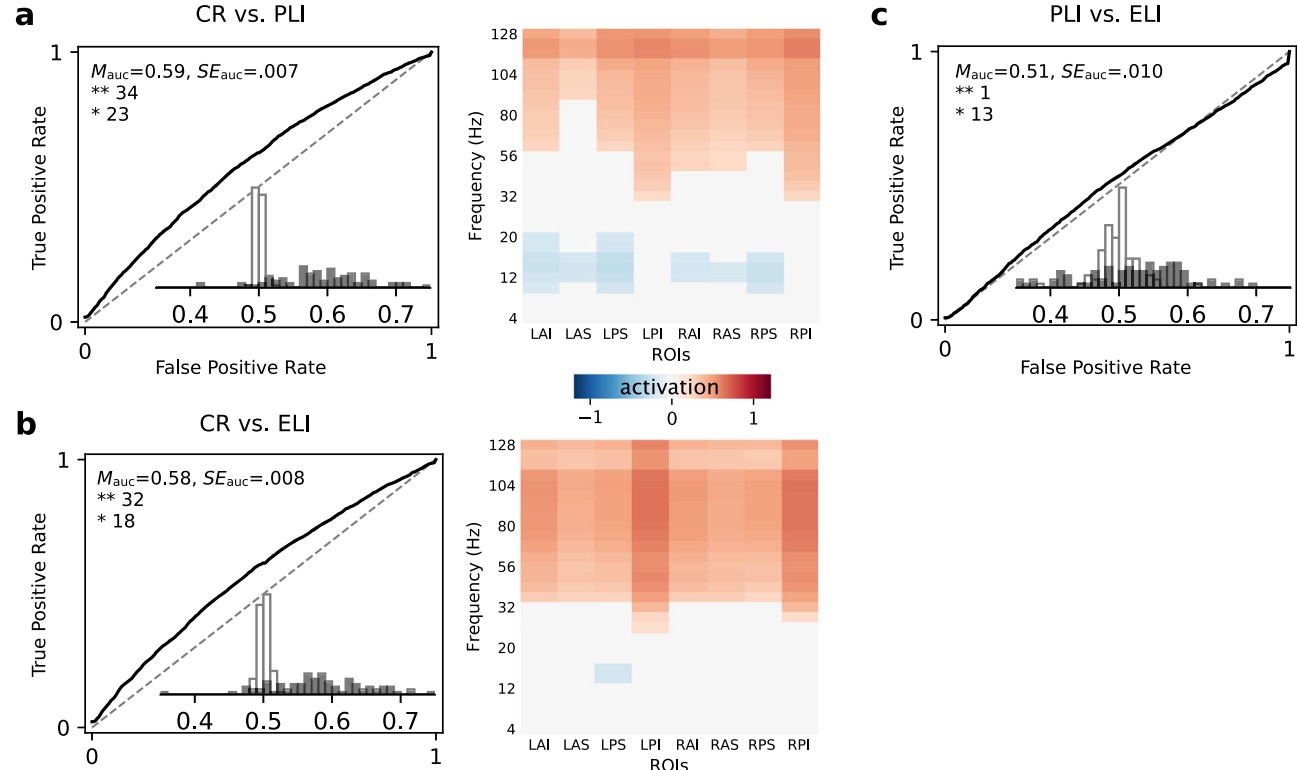

**Fig. 5 | Performance and feature activation of recall-subtype classifiers. a** Correct recall (CR) vs. prior-list intrusion (PLI) classifiers. **b** Correct recall (CR) vs. extra-list intrusion (ELI) classifiers. **c** PLI vs. ELI classifiers. Visualization and notations as in Fig. 4. Source data are provided as a Source Data file.

Figure 6a shows the prevalence of each type of clustered response across serial positions.

When trained under event re-sampling we found a small "primacy bump" in the output probability predicted by the significant encoding classifiers around early list positions, followed by a gradual decline through the rest of the list (Fig. 6b). Moreover, we found a clear separation based on the clustering status of the studied items. We used a linear mixed-effects model (see Methods) to characterize how classifier output differed among the five event subtypes (BC, TC, SC, NC, and non-recalled) at early (1–4), middle (5–20), and late (21–24) list positions. The model revealed a significant main effect of event subtype, $F(4,1204) = 172.08$, $MS_e = 0.025$, $p < 0.001$, and list position, $F(2,1204) = 81.91$, $MS_e = 0.012$, $p < 0.001$, but a non-significant interaction, $F(8,1204) = 1.76$, $MS_e = 0.0003$, $p = 0.08$. Pairwise comparison of the estimated marginal means (EMMs), with Bonferroni correction, showed that BC and TC items both had significantly higher classifier outputs than SC items and NC items (all corrected $p$s < 0.001). Classifier output also declined across the three serial position groups (all corrected $p$s < 0.001).

Classifiers trained on all events declined sharply over the encoding list, consistent with the idea that they may have learned to rely on signals predictive of early versus late items regardless of memory success (Fig. 6c). Applying the same mixed-effects model revealed significant main effects of list position, $F(2,1218) = 388.70$, $MS_e = 0.118$, $p < 0.001$, and event subtype, $F(4,1218) = 90.59$, $MS_e = 0.027$, $p < 0.001$, without a significant interaction, $F(8, 1218) = 1.15$, $MS_e = 0.0003$, n.s. Classifier output was higher among BC and TC items than NC items (corrected $p$s < 0.05). Classifier output for BC items also exceeded that of SC items (corrected $p < 0.001$), but SC items did not reliably differ from TC or NC items.

We conducted a parallel analysis to examine whether the significant retrieval classifiers tracked memory states throughout the recall phase and as a function of temporal and semantic organization.

We partitioned correct recalls based on their relation to the subsequently recalled item into four response subtypes (BC, TC, SC, NC), echoing those in the encoding classifier analyses. Figure 6d shows the probability of each response type as a function of time during the recall period. When trained on re-sampled retrieval events, classifier outputs showed that the pre-recall neural activity that distinguished correct recalls and intrusions did not vary as a function of recall time or transition subtypes (Fig. 6e). The mixed-effects model on the output from the event re-sampling classifiers revealed a marginal effect of recall time (early: first three 7.5 s recall bins, middle: middle four recall bins, late: last three recall bins), $F(2,793.43) = 3.24$, $MS_e = 0.0015$, $p = 0.04$. A significant main effect of event subtype (including BC, TC, SC, NC, PLI, and ELI), $F(5,793.36) = 76.49$, $MS_e = 0.036$, $p < 0.001$, primarily reflected the large difference between classifier outputs associated with correct recalls and intrusions, with no reliable differences across the cluster or intrusion subtypes (model-based EMM comparisons, Bonferroni corrected). We also found an interaction between recall time and event subtype, $F(10,793.35) = 2.20$, $MS_e = 0.001$, $p < 0.05$, driven by a separation between NC and SC items (corrected $p < 0.01$) and a separation between PLI and ELI items (corrected $p < 0.001$) towards the end of the recall period.

When trained on all events, classifier output exhibited a strong monotonic decline over the 75 s recall period (Fig. 6f). The mixed-effects model of these classifier outputs revealed a significant main effect of recall time, $F(2,1061.7) = 43.61$, $MS_e = 0.032$, $p < 0.001$, and event subtype, $F(5,1061.6) = 83.31$, $MS_e = 0.061$, $p < 0.001$, but no significant interaction. Classifier output decreased significantly from early recalls to middle recalls (corrected $p < 0.001$), but did not reliably change into the late recall interval. Classifier outputs for PLIs and ELIs were significantly lower than for all other event subtypes (all corrected $p$s < 0.001), but the other event subtypes did not differ reliably from one another.

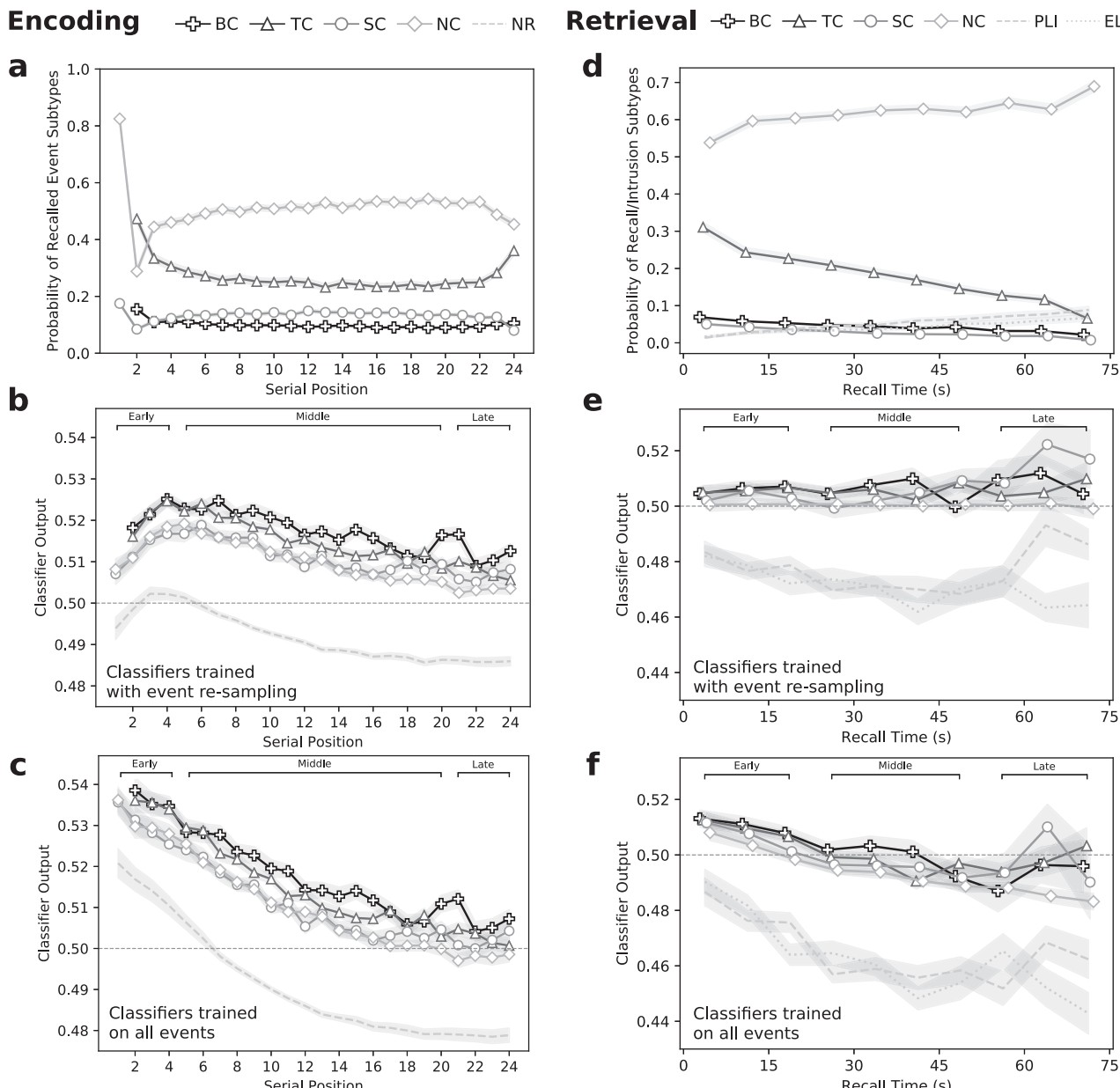

**Fig. 6 | Classifier dynamics and recall organization. a–c** Encoding classifier dynamics for subsequently recalled and non-recalled (NR) words, with recalled words partitioned into the following event types: both temporally and semantically clustered (BC), temporally clustered only (TC), semantically clustered only (SC) and non-clustered (NC). **a** shows the probability of each recalled event subtype as a function of serial position. **b, c** illustrate classifier output for encoding classifiers trained with and without event re-sampling, aggregated across all participants with statistically significant classifiers. **d–f** Retrieval classifier dynamics for event types that vary in clustering status (BC, TC, SC, NC) and for prior-list and extra-list intrusions (PLIs and ELIs) across the 75 s recall period. **d** illustrates the probability of events over binned recall time, and (**e, f**) illustrate the output of retrieval classifiers trained with and without event re-sampling. Shaded regions represent within-participant standard errors[45]. Source data are provided as a Source Data file.

## Predicting individual differences in temporal and semantic clustering

Encoding classifiers, which were only trained to predict the binary labels of subsequent recall success, nonetheless discriminated recalled items based on their subsequent temporal and semantic clustering (Fig. 6b, c). As this sensitivity appeared in the aggregated classifier output from all significant classifiers, we asked whether the classifiers also predict participant-level variability in subsequent clustering as indexed by behavioral scores of temporal and semantic clustering[4], using a set of corresponding classifier-based neural scores (see Methods).

Figure 7 illustrates the correlations between classifier-based neural scores and participants' behavioral temporal and semantic clustering scores. Participants whose classifier output probabilities differed between temporally- or semantically-clustered items and non-clustered items generally exhibited stronger behavioral effects of temporal and semantic clustering. When training classifiers with event re-sampling, which controls for serial position effects, we found statistically significant correlations between behavioral and neural temporal clustering (Fig. 7a, $r(86) = 0.34$, $p = 0.001$) and between behavioral and neural semantic clustering (Fig. 7b, $r(86) = 0.24$, $p = 0.027$).

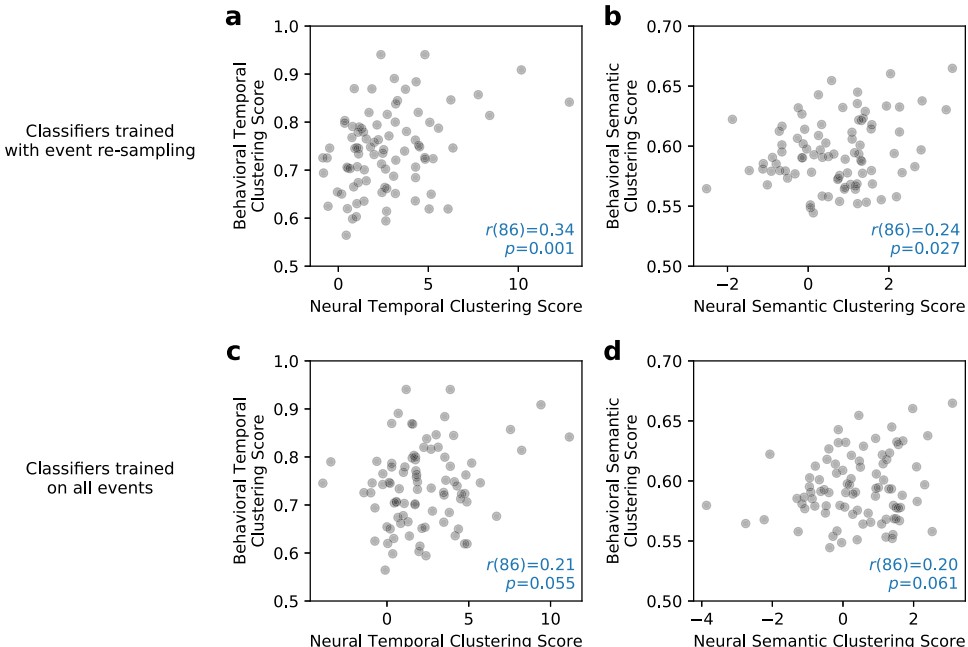

**Fig. 7 | Correlating neural and behavioral indices of temporal and semantic clustering.** We constructed neural indices of clustering by conducting two-sided independent *t*-tests on the classifier outputs associated with subsequently-clustered and non-clustered recalled items for each individual. Comparing these neural indices with behavioral clustering measures using two-sided Pearson correlation revealed a significant positive relation for both temporal and semantic clustering for classifiers trained under event re-sampling (**a**, **b**). For classifiers trained on all events, we observed weaker correlations that trended in the same direction (**c**, **d**). Source data are provided as a Source Data file.

Figure 7c, d illustrate the correlations obtained using classifiers trained with all events. Here we also observed positive correlations between behavioral and neural clustering measures, but these correlations fell short of our statistical significance threshold (temporal clustering: $r(86) = 0.21$, $p = 0.055$; semantic clustering: $r(86) = 0.20$, $p = 0.061$). Nonetheless, our two classification approaches did not lead to reliable differences in the correlations between behavioral and neural clustering measures: The mean difference between bootstrapped Pearson's *r*s was −0.08 (95% CI [−0.19, 0.05]) for the temporal clustering correlations, and −0.07, 95% CI [−0.16, 0.02] for the semantic clustering correlations. Future work may provide better tracking of variability in temporal- or semantic-driven encoding processes by building multi-class decoders to directly optimize the prediction of neural responses associated with different event clustering subtypes.

## Discussion

We show that models based on spectral EEG features can reliably distinguish between the encoding of subsequently recalled and forgotten items and between the imminent recall of correct list items and intrusions. Consistent with univariate analyses, these individual-participant classifiers revealed elevated high-frequency (gamma) activity and suppressed alpha activity as spectral signatures of good memory (Fig. 4). Our memory encoding classification results also align with prior intracranial EEG findings[12,13,16,33]. In the case of retrieval, however, our study demonstrates participant-specific classification of both prior-list and extra-list intrusions (Fig. 5). The features recovered by our retrieval classifiers match findings from univariate analyses of memory intrusions in a study using indwelling electrodes in the human hippocampus[10]. Both studies show that increased HFA signals correct responses, with our study showing that such signals can be observed non-invasively, and that decoders can reliably use these signals to predict retrieval success in individual participants. In demonstrating these results, we leveraged our unusual design in which each participant contributed data from 576 lists across 24 experimental sessions.

As noted by one previous study[34], variability in memory performance across items can reflect a combination of underlying factors, including endogenous fluctuations in brain states that support successful memory formation and exogenous factors that influence memory performance via other channels. We know, for example, that in delayed free recall, participants exhibit far better recall for early list items than for items from latter list positions (the well-established law of primacy[32,35]). We highlight that the event re-sampling technique was key to allowing us to determine whether our scalp-EEG decoders were solely picking up on list position-related differences in recall, where we re-sampled training events to ensure equal numbers of correctly recalled items in each list position before training the classifiers. We found that these decoders achieved similar overall levels of classification performance to decoders trained on all events, but they showed a reduced reliance on high-frequency neural activity (Fig. 4a, b).

To successfully remember which words occurred on a particular list, participants must form and retrieve associations between items and their situational or list context. Experimental psychology has extensively documented the behavioral evidence for such associations as seen in participants' tendency to subsequently recall items clustered according to their temporal and semantic associations with other list items. To the extent that our classifiers identify features underlying contextual associations, we would expect to find higher classifier outputs during the encoding and prior to the recall of subsequently clustered items. We found that encoding classifiers predicted increased recall probability for subsequently clustered items, an effect that appeared regardless of whether the classifiers were trained on all events or on re-sampled events. We did not, however, find an analogous clustering effect for our retrieval classifiers.

Unlike group-level analyses of neural data, decoders provide a participant-specific model relating neural activity to behavior. We leveraged these models by asking whether individual differences in the degree to which people clustered their recalls correlated with

differences in the degree to which their encoding classifiers distinguished clustered and non-clustered items. In the case of classifiers trained on re-sampled events—which removed effects of list position from the model—we observed a striking positive correlation between neural and behavioral indices of temporal and spatial clustering. These findings indicate that the classification-based model of encoding success includes at least some features that reflect memory-specific contextual-encoding processes. The matched event distributions from the re-sampling technique enabled the classifiers to better pick up these nuanced signals. The ability of healthy participants to each contribute data of hundreds of lists with high recall rates enabled us to detect these relations between brain and behavior.

The free recall task is ideally suited to observe the neural mechanisms underlying conscious recollection; here, participants attempt to remember the items from a previously experienced list, with list context serving as the retrieval cue. When they commit recall errors they typically believe that they have correctly recalled previously studied items[36]. Thus, the neural activity that differentiates these classes of responses provides a specific signal related to the veridicality of the retrieved information. That such signals appear in scalp-recorded HFA offers promise for future non-invasive approaches to measuring, and perhaps modifying, the neural substrates of conscious recollection.

## Methods

### Participants

The data reported here come from Experiment 4 of the Penn Electrophysiology of Encoding and Retrieval Study (PEERS). Prior studies[34,37] reported detailed behavioral analyses of this study. Here we focus on the electrophysiological correlates of encoding and recall behavior. Ninety-eight young adults (52 female, mean age = 21.45, $SD = 3.06$), recruited from among the students and staff at the University of Pennsylvania and neighboring institutions, each contributed 24 sessions of multi-list delayed free recall data. The experimental protocol was approved by the Institutional Review Board at the University of Pennsylvania, and all participants provided informed consent. Monetary compensation was provided to participants for each session completed, with bonus payment rewarded for minimizing eyeblink rates during stimulus presentation, maximizing performance during distractor tasks, and for completion of the 24th session. To ensure adequate representation of successful and unsuccessful mnemonic events we decided, prior to carrying out any analyses, to only include participants with recall rates between 15% and 85% for 10 or more sessions. All reported analyses were based on the 88 participants who met this criteria. No gender-based analyses were carried out during this study.

### Experimental task

In each of the 24 experimental sessions participants completed 24 lists of a delayed free recall task. In each list, participants first studied 24 session-unique English words. Words in each list were drawn without replacement from a pool of 576 common English words (see ref. 34, for details regarding list construction). Each word appeared individually onscreen for 1600 ms, and was followed by an interstimulus interval of 800–1200 ms (uniformly distributed). Following list presentation, participants performed a distractor task for 24 s. The distractor task consisted of answering math problems of the form $A + B + C = ?$, where $A$, $B$, and $C$ were positive, single-digit integers, though the answer could have been one or two digits. When a math problem was presented on the screen, participants typed the sum as quickly and accurately as possible. After the post-encoding distractor task, there was a jittered delay of 1200–1400 ms, after which a tone sounded, a row of asterisks appeared, and the participant was given 75 s to freely recall the studied items. Participants were given a short break (about 3 min) after every eighth list in each session.

### Data collection and pre-processing

We recorded EEG with either a 129-channel EGI Geodesic Sensor Net ($N = 53$) in the Netstation acquisition environment or with a 128-channel BioSemi Active Two system ($N = 35$). Within each participant, we used the same system to record all 24 sessions. We applied a 0.1 Hz high-pass filter to remove the baseline drift of the EEG signals over the course of each session. In addition, we applied a fourth-order Butterworth notch filter with a 58–62 Hz stop-band to attenuate electrical line noise.

We re-referenced recordings to the common average of all electrodes. When calculating the voltage mean across all electrodes, we excluded bad electrodes based on the channel variance and Hurst exponent. Specifically, raw session recordings were high-pass filtered at 0.5 Hz to reduce the impact of baseline drift on the variance and Hurst exponent. We then partitioned the EEG recording into three sections, separated by the two mid-session breaks. Within each partition, we calculated the log-variance and Hurst exponent of each (non-electrooculogram) channel and z-scored across channels. We marked channels as bad if their z-scored log-variance exceeded +3.0 or fell below −3.0, or if their z-scored Hurst exponent exceeded 3.0 during any of the three partitions.

### Event and feature construction

**Spectral decomposition.** We used the multitaper method implemented in the MNE Python software package[38,39] (Version 0.15) to estimate spectral power at each electrode over 4–128 Hz. We spaced frequencies every 2 Hz in the range of 4–26 Hz, and every 6 Hz within 26–128 Hz, resulting in 29 frequencies of interest. We used a 500 ms moving window centered at multiple time-points relative to the start of word encoding events and free recall vocalization events, with a 50 ms step size. We chose the multitaper method over the Morlet wavelet method, as convolving low frequency wavelets using buffer periods may allow speech artifacts to intrude in the power estimates of intervals just prior to participants' vocal responses. Within each session, we log-transformed and z-scored power across events at each frequency and electrode pair, separately for encoding or retrieval events.

**Epoch construction.** We partitioned encoding or retrieval trials into two event classes. During the encoding of a list, we defined successful memory events as words that were subsequently recalled within the corresponding recall period of the list, and unsuccessful memory events as words that were not recalled. During retrieval, we defined successful memory events as the memory search intervals immediately preceding recalls of list items (correct recalls). Unsuccessful memory search periods preceded recall errors, including recall of words from previous lists (prior-list intrusions, PLIs) and from outside of the word pool (extra-list intrusions, ELIs). To minimize speech artifacts, we excluded recall responses which began within 1 s after the onset of the previous recall. For the encoding events, we averaged the spectral patterns from 250 ms to 1350 ms after word onset, effectively covering the 0–1600 ms presentation window. For memory search intervals in the free recall phases, we extracted the spectral patterns at 250 ms prior to speech onset, effectively covering a 500 ms memory search window before the observed onset of recall responses.

**Event re-sampling.** We re-sampled the training events to achieve an equal ratio of positive and negative instances across serial positions (for the encoding classifier) and recall bins (for the retrieval classifier). For each session, we down-sampled events at each serial position so that the ratio of positive and negative events at that serial position matches the session-level ratio across all serial positions. Because recall responses occur at variable time-points throughout the 75 s free recall period, we first binned the recall responses within every 7.5 s, and down-sampled either the correct recalls or intrusions within each bin to approximate the session-level global rate of recall accuracy. On

average, re-sampling the training events within-individual preserved 76.7% of the encoding events and 53.2% of the retrieval events.

**Event clustering type.** During classifier evaluation, we further partitioned the successful memory events into four sub-types: (1) BC, both temporally and semantically clustered, (2) TC, temporally clustered but not semantically clustered, (3) SC, semantically clustered but not temporally clustered, and (4) NC, non-clustered (i.e., temporally and semantically isolated). During encoding, we labeled each remembered word based on the relationship between its corresponding recall response and the neighboring recall responses. A subsequently temporally clustered encoding event (with list serial position $i$) has a corresponding recall response either immediately preceding or immediately following recall of the previous list item (serial position $i - 1$). A subsequently semantically clustered encoding event has a corresponding recall response whose similarity score with either the previous or the next recall response, calculated by the cosine similarity of the word embeddings in the word association space, is ≥0.4 (see also refs. 30, 40). During retrieval, we labeled each correct recall event based on its temporal and semantic relationships with the next recall event. A temporally clustered recall event preceded its encoding list neighbors, with an absolute transition lag (i.e., the difference between their serial positions) of 1. A semantically clustered recall event preceded the recall of a semantically related item, defined as having a word-association score ≥0.4.

### Decoding successful memory

**Univariate analyses.** We collapsed spectral power estimates across subsets of electrodes to generate average spectral patterns at 29 frequencies × 8 regions of interest (ROIs). We selected ROIs based on previous studies using similar EEG caps[30,41]. An independent $t$-test, performed separately on each participant's data, assessed the spectral difference between successful and unsuccessful memory events at each frequency-ROI pair. A one sample $t$-test on the resulting (across participant) distribution of $t$-statistics evaluated the group-level effects.

**Multivariate classification.** We trained participant-specific, L2-penalized logistic regression classifiers to decode brain states associated with successful encoding or retrieval events, using power at each frequency-electrode pair as input features. Classifiers were trained using a leave-one-session-out cross-validation procedure on sessions 1 to 23. For classifiers trained with event re-sampling, we conducted 20 runs of the cross-validated classification procedure with independent rounds of training event re-sampling, and tested on the full set of encoding or retrieval events. The baseline model was trained and tested on all events in one round of the cross-validation procedure. The reported analyses included an average of 22.3 sessions for each of the 88 participants. We excluded features from electrodes close to the face and the neck (27 excluded for Geodesic caps, 18 excluded for BioSemi caps) to minimize electromyographic artifacts.

In each cross-validation fold, we trained the classifier by minimizing the following loss function over the set of weights, $w$, and $n$ training events:

$$\min_{\mathbf{w}} \frac{1}{2} \mathbf{w}^T \mathbf{w} + C \sum_{i=1}^{n} \log(\exp(-y_i (\mathbf{X}_i^T \mathbf{w})) + 1) \quad (1)$$

$X_i$ is a set of frequency × electrode features for a given encoding or retrieval event, and $y_i$ is the corresponding event class label indicating memory success. The hyperparameter search around the best inverse regularization parameter, $C$, was conducted separately across the two training distributions (with event-sampling or all events) and for the two recording systems. $C = 2.15^{-5}$ from 10 log-spaced values from $10^{-8}$ to $10^{-1}$ optimized the classification accuracy of encoding success on the aggregated 24th session data from all participants with a leave-one-

participant-out cross-validation procedure in all four cases, and was subsequently shared across all cross-validation folds when training both the encoding and the retrieval classifiers. To adjust for class imbalance, we weighted observations inversely proportional to their class frequencies.

**Classifier evaluation.** We evaluated classifier performance by computing the area under the receiver operating characteristic curve (i.e., AUC) of the predicted probabilities of held-out test events across all folds. To determine whether each classifier performed above chance, we obtained a distribution of 100 baseline AUC scores for each individual classifier by repeating the cross-validation training procedure on permuted data, where the event class labels within each training session for a given fold were randomly reassigned to events within that session. The event class labels in the held-out test session in each fold remained unshuffled. A significant classifier must have an observed AUC above 95% of the baseline AUC distribution. At the group level, we used a paired $t$-test to determine whether the distribution of the observed AUCs was significantly higher than the distribution of individual mean baseline AUCs.

We also derived feature activation maps by constructing forward models from the trained classifier weights, similar to prior work[12,13,28]. For each individual classifier, we first averaged the activation values at each frequency-electrode pair across all training folds, then pooled the activation values across electrodes within each ROI. At the group level, we performed a one-sample $t$-test on the mean activation values at each frequency-ROI pair.

### Classifier-based analysis of memory dynamics

We constructed linear mixed-effects models[42] to assess how classifier output probabilities on held-out data tracked event clustering types (BC, TC, SC, NC, and the unsuccessful memory events) over three item position ranges (early, middle, and late). We defined the first four list items as "early", the last four list items as "late", and the remaining items as "middle"[21]. During retrieval, the responses within the first three recall bins (each spanning 7.5 s) and the last three recall bins were designated as the early and late responses. This analysis only included predictions from the significant classifiers, with each classifier contributing a mean output probability for each event sub-type × item position pair. For both the encoding and retrieval mixed effects models, we included event type, item list position, and the interaction between them as fixed effects, with participant-level random intercepts. We tested hypotheses using a Type III ANOVA with degrees of freedom approximated by the Satterthwaite method[43], as implemented in the lmerTest package for R[44].

To quantify the relation between classifier-based neural indices and behavioral indices of memory dynamics we computed the Pearson correlation, across participants, of these two measures. We computed participant-specific temporal and semantic clustering scores based on a previous study[4], as implemented in https://github.com/pennmem/pybeh. We derived neural temporal clustering scores and semantic clustering scores as the $t$-statistics of independent $t$-tests between the classifier output of (1) TC items and NC items, and (2) SC items and NC items, respectively.

### Reporting summary

Further information on research design is available in the Nature Portfolio Reporting Summary linked to this article.

## Data availability

The raw EEG data generated in this study have been published in BIDS format on the OpenNeuro platform as experiment PEERS4 within dataset ds004395 and have been made publicly available at https://doi.org/10.18112/openneuro.ds004395.v2.0.0. The source data for all figures are provided in the Source Data files. Source data are provided with this paper.

## Code availability

All analysis code has been made publicly available at https://github.com/pennmem/EEG-memory-dynamics-public.

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

## Acknowledgements

We gratefully acknowledge support from the National Institute of Mental Health grant MH55687 to M.J.K. Correspondence concerning this study may be addressed to M.J.K. (kahana@psych.upenn.edu), Y.L. (liyuxuan@stanford.edu), or J.K.P. (pazderaj@mcmaster.ca).

## Author contributions

Y.L.: conceptualization, methodology, software, formal analysis, investigation, data curation, writing - original draft, writing - review & editing, visualization; J.K.P.: methodology, software, data curation, writing - original draft, writing - review & editing; M.J.K.: conceptualization, methodology, resources, writing - original draft, writing - review & editing, supervision, project administration, funding acquisition.

## Competing interests

The authors declare no competing interests.
