## [Peer Review File · Nature Communications]

EEG Decoders Track Memory DynamicsREVIEWER COMMENTS

Reviewer #1 (Remarks to the Author):

Summary:

Pazdera and colleagues have used scalp electroencephalography to investigate memory in almost 100 subjects with each 24 sessions using a delayed recall task. Using conventional power analyses and logistic regression classifiers they replicated findings in showing that successful encoding is characterized by an increase in HFA and a decrease in LFA in most, as well as an increase in theta power in some brain areas. Similarly, HFA and LFA before vocalization predicted whether a vocalized word was previously learned. Moreover, they demonstrate that the neural correlates that predict memory success decline across list positions, showing primacy effects. This latter finding replicates to some extent previous findings from the same group (i.e. Sederberg et al., 2006; Neuroimage).

General Comments

First, I want to applaud the authors on the herculean effort that went into this project. Collecting data from almost 100 participants, with each participating in multiple sessions is impressive. Second, the fact that results from invasive EEG recordings could be replicated with non-invasive EEG recordings is reassuring and line with previous non-invasive work. I therefore think that this article will be relevant for the community. Having said this, I feel that this paper is mostly replicating previous findings, which have shown similar findings also with non-invasive electrophysiology. I am therefore not sure about the conceptual advance that is offered by this study (see major points 5a and b).

Major Points

1. It is intriguing that the classifier over the list position tracks the primacy effect. The forward model of the classifier showed a reliance on specific spectral features (i.e., lower LFA and higher HFA for hits). These two findings combined shed doubt on the claim that lower LFA and higher HFA causally relate to memory formation in an absolute way. This becomes apparent when comparing the encoding classifier outcomes at serial position seven for subsequently forgotten and serial position 17 for subsequently remembered words. Why does the same spectral environment lead to different outcomes (later forgotten

vs. remembered) at different times of list learning? Could it be, that the primacy effect, which by definition overlaps with an increased memory performance for earlier learned words, is characterized by a low LFA and high HFA, but later words are remembered using a different route?

2. As a first step it would be very interesting whether a decreased LFA and increased HFA act as the differentiator between subsequently forgotten and remembered trials over all list positions. For example, it is conceivable that subsequently remembered words in the beginning are characterized by their HFA, but later ones are not. This is something the authors can easily address by performing classifiers per list position and comparing their forward models. Likewise, it would be very interesting to see the time-frequency resolved activity over the full 24-word list separate for subsequently forgotten and remembered words. As it is, the paper lacks a fleshed-out discussion of why the relative and not absolute power difference tracks subsequently forgotten or remembered words.

3. The authors report an increase of HFA and decrease of LFA and report this as a spectral tilt for to be remembered items. It needs to be verified whether these results stem from an aperiodic spectral tilt or reflect oscillatory changes (see Donoghue et al., 2020, Nat. Neur.). This is also relevant in light of the results reported by Fellner et al (2019; PLoS Biol), who show that low and high frequency correlates of successful memory encoding are dissociated, therefore speaking against a pure tilt based view of electrophysiological correlates of memory success.

4. Why is the minimum time between two answers at recall measured from the response beginning, instead of its end? Only the latter would account for variability in word vocalization.

5a. Scholarship. Out of the 46 references, 32 (70%) come from the senior author's group. I understand that the senior author is a leading figure in the field, but other groups too have published on memory effects using electrophysiology and have ascribed slightly different roles for low frequency power decreases and high frequency power increases. I think that an effort should be made in the introduction and discussion section to give a more balanced overview of the literature and to better embed the relevance of their findings in the field.

5b. This point is related to the above. Page 2, 4th paragraph "We began this project unsure of whether scalp EEG possesses the requisite fidelity to decode variability in mnemonic success, both during encoding and retrieval. Although prior intracranial EEG work suggested

a positive outcome, the paucity of similar demonstrations using non-invasive recordings made us question whether scalp EEG data could support reliable trial-level classification.” I don’t think this statement is justified given the plethora of papers that demonstrate subsequent memory effects and memory recall effects using non-invasive electrophysiological methods. Here are a few examples of empirical papers (especially the first two seem very relevant here):

<https://pubmed.ncbi.nlm.nih.gov/30042664/>

<https://pubmed.ncbi.nlm.nih.gov/24064073/>

<https://pubmed.ncbi.nlm.nih.gov/19001457/>

<https://pubmed.ncbi.nlm.nih.gov/16837600/>

<https://pubmed.ncbi.nlm.nih.gov/21162031/>

<https://pubmed.ncbi.nlm.nih.gov/17335397/>

<https://pubmed.ncbi.nlm.nih.gov/19922772/>

<https://pubmed.ncbi.nlm.nih.gov/24632089/>

Here are two review papers:

<https://pubmed.ncbi.nlm.nih.gov/20060015/> (cf Table 1)

<https://pubmed.ncbi.nlm.nih.gov/23769913/>

Minor points

a. page 2, 3rd paragraph “Non-invasive EEG studies have found greater spectral tilt for successful as compared with unsuccessful memory encoding (Long et al., 2014; Long & Kuhl, 2019; Sederberg et

al., 2006).” It took me a while to understand that greater tilt here means a counter clockwise rotation (i.e. low frequency power decreases and high frequency power increases). I always thought of it the other way around, and other readers might be similarly confused. It would be good to specify what the authors mean with greater/smaller tilt right from the get-go to avoid this confusion.

b. page 3, Participants. “ ... we excluded 10 subjects prior to analysis whose recall rates were between 15% and 85% for fewer than 10 sessions.” I did not understand this statement. Did you exclude subjects who were performing exceedingly poor or exceedingly good in at least 10 sessions? Please specify.

c. page 4, Spectral Decomposition. “We log-transformed and normalized power across all events from the same task phase (encoding or retrieval) within each session.” Please specify exactly how power was normalized, ideally by providing a formula.

d. Please clarify that the classifier output is label prediction and not prediction accuracy.

e. Figure 4A and 4B the scatter plot is very hard to see. Perhaps reducing the transparency and scattering the individual datapoints in a way so they do not overlap would yield a better result. A distribution next to it would work as well. Currently the error bars and mean obstruct the view of the scatter plot.

f. What is the advantage of using the L2-norm in the cost function? It is more susceptible to outliers compared to the L1-norm.

g. I appreciate that the authors have published other papers where they investigate the behavioural data of this same dataset in more detail. But I would still like to see some numbers here regarding hits/misses, especially over time. This is also important when talking about the cost function and how class imbalance is counteracted by weighting the classes. The cost function includes a weight to balance out uneven trial numbers for example if there are more correctly remembered trials than prior list intrusions. How did you make sure that there are enough trials in each session to have a valid representation of the event? Is there an increase of PLI and ELI within a list and over lists? At least the former would be expected independent of attentional decrease. If there is one, how does such a bias effect the results?

h. How long were the “short breaks” every eight lists?

i. At one point the authors use a linear mixed effects model on the classifier performance to distinguish event types at different list positions. For the retrieval period the first and last 22.5 seconds were chosen as the early and late list positions. A close look at Figure 3A reveals that participants might not have a single response in the late period, whereas the majority of responses are given during the early period.

j. The abstract states that 98 participants performed the experiment. This includes the excluded participants. This should either be adapted in the abstract or made clear under Methods-> Participants that all subsequent analyses are based on 88 participants, which is still an impressive sample size.

k. In Figure 2A why does Subject 8 have a uniform difference power spectrum? Is this the

color scaling?

I. Page 14, line 6. "In particular, classifier output was higher among BC ($M = 0.520$, $SE = 0.001$) and TC ($M = 0.517$, $SE = 0.001$) items compared to NC items ($M = 0.513$, $SE = 0.001$)."
<- significantly higher?

Reviewer #2 (Remarks to the Author):

The current study investigates EEG correlates of successful versus unsuccessful memory encoding and recall. The authors first characterize shifts in spectral power that occur during word encoding and preceding recall. They then use a pattern classification approach to test whether the observed correlates are sufficiently reliable to predict memory for individual trials on held out sessions. They find evidence for a spectral "tilt" whereby high frequency power is increased and low frequency power is decreased (with the exception of theta) for successful versus unsuccessful memory encoding and recall. Encoding and recall classification is generally very good, although different types of recall errors (prior vs extra list intrusions) cannot be reliably discriminated. Classifier evidence for successful memory also showed a primacy gradient and higher levels during encoding of words that would later be clustered by temporal and/or semantic context.

Overall, the paper is very well written and I find the results to be compelling (with some exceptions described below). Although there is no obvious theoretical advance here, the strengths of the paper lie in the "big data" approach. The study massively sampled both sessions and subjects resulting in remarkable statistical power that enables analyses that are not feasible with more traditional datasets. I think the approach moves the field forward, and although I would like to see more theoretically exciting questions addressed here, this is an important first step.

Major concerns:

My main concern is with the serial position and clustering analyses, which I find the most theoretically interesting. In particular, I'm concerned that the pattern of results could emerge based on bias in the training data. For example, if more training data is coming from

earlier serial positions for recalled compared to not recalled, that could result in the primacy gradient observed simply because the classifier confounds subsequent memory with serial position. The same logic holds for retrieval and for the clustering analyses depending on the frequencies of BC, TC, etc. (also see comment below). To address this issue, the same permutation-based testing approach should be taken in this set of analyses as in the preceding analyses, which should address any potential bias.

Some description of the behavioral data is essential to include here. In particular, in order to interpret the neural data, we need to see frequency of PLI and ELI (a quick skim of the previous papers suggests this has yet to be described). Serial position curves mirroring Fig 6 and some characterization of temporal and semantic clustering is also needed.

Classifier evaluation:

The within-subject permutation test is a strong test of classifier performance but the resulting null distributions should be shown, at least for the three example subjects. The binomial test for the group level data is less clear. Is each classifier (fold) treated independently? If so, nonindependence within subjects would violate key assumptions of the binomial distribution. I recommend at least one additional confirmatory analysis -- a simple group-level t-test comparing the actual and permutation-derived chance AUC values across subjects.

Minor concerns/clarifications:

Classification of correct recall versus PLI and ELI is a subset of correct versus incorrect recall. The nonindependence of these analyses should be made more clear.

Is the increase in HFA really concurrent with the decrease in LFA? This claim should be supported by a time-frequency analysis.

How sensitive is classifier performance to the choice of regularization parameter? Why was the same parameter used for encoding and recall classifiers, and why was it based on classifying encoding states specifically? This choice might have influenced the observation

that performance was somewhat more reliable for encoding compared to recall.

The description of the encoding task is a little unclear. What were the instructions provided? If no response had to be made, was there anything done to ensure that the words were processed (e.g., eye tracking?)

In the intro, the authors suggest that reliable single-trial classification of memory states in scalp EEG data is quite limited. Although that seems largely true in episodic memory, single trial decoding is quite commonly demonstrated in the working memory literature. Thus, I would recommend qualifying that statement.

Describing EEG memory contrasts as biomarkers seems premature. Here the goal is simply to characterize the SME and recall effects whereas the term biomarker is typically used in a translational context to describe something with diagnostic relevance.

Regarding the theta band results, the line, “supporting other recent studies linking lower theta band activity...with mnemonic success” is confusing. Does lower here refer to lower frequency, since the SME is actually associated with a power increase in theta?

Reviewer #3 (Remarks to the Author):

In this manuscript, authors report decodability of scalp EEG signals for successful mnemonic events. Univariate analyses replicated the spectral tilt associated with successful encoding and retrieval, which has been shown in intracranial EEG studies from this group. Classifiers trained using power across frequency bands and electrodes as features reliably predicted subsequent recall from encoding data, and correct recall from signals prior to recall.

Forward model constructed based on the classifiers revealed feature activation maps that resembled spectral tilts in univariate analysis. Encoding classifier outputs were stronger for early items in the word lists and for items that were recalled with temporal and semantic clustering. The findings that classifier outputs mirrored two very-well established, strong behavioral effects (primacy and clustering/item-context binding) provides good evidence that the classifiers were able to decode neural signals that underlie the exact mnemonic

processes.

The study is theoretically well-grounded, and conducted with clear hypotheses and solid methods. The manuscript is also very well-written, and I only have a few suggestions and clarification questions for the authors.

1. Classifier outputs for recall activity were below chance level except for recalls that were made early in the retrieval phase (Figure 6B). Given the approaches taken, I'm having a hard time wrapping my head around why that would be the case. It would be worth to bring in the authors' take on this in the Discussion.

2. The following descriptions need to be clarified.

a. "we converted recordings to the average reference" This sounds like recordings were replaced with average reference, whereas authors probably meant that "recordings were off-line referenced to common average signals."

b. "we applied a fourth-order 4 Hz Butterworth notch filter at 60 Hz to attenuate electrical line noise." It is unclear to me how a 4Hz filter can be applied at 60Hz – please clarify.

c. "We excluded peripheral electrodes (27 excluded for Geodesic caps, 18 excluded for BioSemi caps) close to the face and the neck in each system during all analyses." Please provide specific criteria for peripheral electrodes (ideally with montages for the two systems), and justify using different numbers of electrodes for data from the two systems. I do not think this would have systematically influenced the results in any way, but as the number of electrodes impacts the number of classifier features and the consequent classifier performance, it would be important to spell out the details for transparency (it can go to appendix).

d. "For memory search intervals in the free recall phases, we extracted the spectral patterns at 250 ms prior to speech onset, effectively covering a 500 ms memory search window before the observed onset of recall responses." I'm confused about this. Was it the instantaneous spectral patterns (from precisely 250ms prior to recall) that were used here? If so, how does that 'cover' 500ms memory search window? It would fall in somewhere in the search window. Maybe the point being made here is that spectral patterns at 250ms

prior to recall should be representative of the signals from the 500ms search window – but that is not clear.

Response Letter (Manuscript NCOMMS-21-15247-T)

We greatly appreciate the thoughtful and constructive comments which we have carefully addressed in the attached revision. Before presenting point-by-point responses to each of the specific comments, we highlight a very substantial change in our analytic framework that addresses a confound noted by several reviewers. Specifically, the reviewers raised the possibility that our encoding classifiers may have learned neural features associated with list position and thus indirectly correlated with mnemonic success. Because serial position predicts recall success, any brain activity that differs across list positions could forecast subsequent recall of the studied items.

To address this concern we present new analyses using an event re-sampling method that ensures equal distributions of recalled and non-recalled items across all list positions during encoding and across all time periods during retrieval (see the new event re-sampling subsection in *Methods*, page 5). Our re-sampling approach combines the best aspects of each reviewer's specific suggestions, but our *cover letter* also reports results from the more specific analyses proposed by the reviewers, which aligned with the new results included in our revision.

Although training classifiers with event resampling provides a more direct test of the theoretical questions posed in our paper, we suspect readers will still want to see the results derived from the more standard classification approach of training on all events. Our revision therefore reports and compares both sets of results. When evaluated on the same set of un-sampled, held-out test events, the decoding performance of the new classifiers trained with re-sampled events closely match that of the original ones (Figures 4A, B). Nonetheless, the new re-sampling method revealed key features of our unique dataset:

- 1) Without event-resampling, univariate and multivariate-classifier analyses revealed increased high-frequency activity (HFA) and decreased alpha accompanying the encoding of successfully recalled items. With event-resampling, which equates the distribution of recalled items across list positions, we no longer find HFA increases; rather, we only see a strong decrease in alpha for successfully remembered items. Thus, it appears that serial position effects must underlie the HFA changes seen in typical subsequent memory comparisons.
- 2) Classifiers trained with event re-sampling exhibited greater sensitivity to the subsequent temporal and semantic organization of the recalled items (Figure 6B,C). Additionally, when trained using event re-sampling, differences in classifier output for clustered and non-clustered items strongly predicted inter-subject variability in temporal and semantic clustering during recall (Figure 7).
- 3) Our analyses of EEG correlates of successful recall – quantified as the difference between pre-vocalization periods leading to correct or incorrect recall – revealed similar univariate and multivariate findings when using event-resampling or when analyzing all retrieval events.

As suggested by the reviewers, we have added several behavioral figures and clarified various ambiguous points; we have also improved the scholarship and breadth of our literature review. Finally, we closely replicated results presented in the previous submission using a substantially improved artifact rejection method. We have updated the methods and the results accordingly. We have also open-sourced our analysis code on Github (<https://github.com/pennmem/EEG-memory-dynamics-public>) and posted all of our raw data on [OpenNeuro.org](https://openneuro.org). Below, we respond point-by-point to the reviewers' comments (original comments in black, responses in red).

Reviewer #1 (Remarks to the Author):

Summary: Pazdera and colleagues have used scalp electroencephalography to investigate memory in almost 100 subjects with each 24 sessions using a delayed recall task. Using conventional power analyses and logistic regression classifiers they replicated findings in showing that successful encoding is characterized by an increase in HFA and a decrease in LFA in most, as well as an increase in theta power in some brain areas. Similarly, HFA and LFA before vocalization predicted whether a vocalized word was previously learned. Moreover, they demonstrate that the neural correlates that predict memory success decline across list positions, showing primacy effects. This latter finding replicates to some extent previous findings from the same group (i.e. Sederberg et al., 2006; Neuroimage).

General Comments: First, I want to applaud the authors on the herculean effort that went into this project. Collecting data from almost 100 participants, with each participating in multiple sessions is impressive. Second, the fact that results from invasive EEG recordings could be replicated with non-invasive EEG recordings is reassuring and line with previous non-invasive work. I therefore think that this article will be relevant for the community. Having said this, I feel that this paper is mostly replicating previous findings, which have shown similar findings also with non-invasive electrophysiology. I am therefore not sure about the conceptual advance that is offered by this study (see major points 5a and b).

Major Points

1. It is intriguing that the classifier over the list position tracks the primacy effect. The forward model of the classifier showed a reliance on specific spectral features (i.e., lower LFA and higher HFA for hits). These two findings combined shed doubt on the claim that lower LFA and higher HFA causally relate to memory formation in an absolute way. This becomes apparent when comparing the encoding classifier outcomes at serial position seven for subsequently forgotten and serial position 17 for subsequently remembered words. Why does the same spectral environment lead to different outcomes (later forgotten vs. remembered) at different times of list learning? Could it be, that the primacy effect, which by definition overlaps with an increased memory performance for earlier learned words, is characterized by a low LFA and high HFA, but later words are remembered using a different route?

Thank you for raising this concern. Our new resampling procedure addresses the potential confound that signals related to the primacy effect or non-mnemonic signals that co-vary across the list positions could have contributed to the spectral profile found in the previous analyses. Based on these new results readers can see that whereas LFA decreases underly encoding success after controlling for list position, HFA appears to be related specifically to the primacy effect..

2. As a first step it would be very interesting whether a decreased LFA and increased HFA act as the differentiator between subsequently forgotten and remembered trials over all list positions. For example, it is conceivable that subsequently remembered words in the beginning are characterized by their HFA, but later ones are not. **This is something the authors can easily address by performing classifiers per list position and comparing their forward models.** Likewise, it would be very interesting to see the time-frequency resolved activity over the full 24-word list separate for subsequently forgotten and remembered words. As it is, the paper lacks a fleshed-out discussion of why the relative and not absolute power difference tracks subsequently forgotten or remembered words.

We appreciate this excellent suggestion and have conducted the reviewer's recommended analysis (training classifiers per list positions and examining the corresponding forward models). Results of this analysis align closely with our resampling results in showing that decreased LFA mark successful encoding at all list positions (see Figure 1, below). We also present, in Figure 2, univariate spectral contrasts for subsequently recalled and non-recalled encoding events separately for each serial position, which shows the same spectral profile.

Figure 1. Forward-model activation maps for encoding classifiers trained separately with data from each serial position.

Figure 2. Univariate differences in spectral power associated with subsequently recalled and non-recalled encoding events at all 24 list serial positions. Z-scored power estimates were aggregated across the eight ROIs.

3. The authors report an increase of HFA and decrease of LFA and report this as a spectral tilt for to be remembered items. It needs to be verified whether these results stem from an aperiodic spectral tilt or reflect oscillatory changes (see Donoghue et al., 2020, Nat. Neur.). This is also relevant in light of the results reported by Fellner et al (2019; PLoS Biol), who show that low and high frequency correlates of successful memory encoding are dissociated, therefore speaking against a pure tilt based view of electrophysiological correlates of memory success.

Our new results are consistent with the view that HFA and LFA reflect distinct processes. We no longer refer to the results as evidence for a spectral tilt and instead discuss the separate contribution of alpha and HFA correlates of memory encoding and retrieval processes.

4. Why is the minimum time between two answers at recall measured from the response beginning, instead of its end? Only the latter would account for variability in word vocalization.

Excellent question! We followed many prior studies that measure latencies between the onsets of successive vocalizations (Murdock & Okada, 1970; Howard & Kahana, 1999; Polyn et al., 2009, etc.). Whereas offset times may prove useful in some circumstances, estimating the time of vocalization offset poses challenges as speakers often draw out their utterances as they approach the end of a word. Even annotating vocalization onsets is a labor-intensive activity, requiring well-trained staff using specialized software (Solway et al., 2010).

5a/b. Scholarship. Out of the 46 references, 32 (70%) come from the senior author's group. I understand that the senior author is a leading figure in the field, but other groups too have published on memory effects using electrophysiology and have ascribed slightly different roles for low frequency power decreases and high frequency power increases. I think that an effort should be made in the introduction and discussion section to give a more balanced overview of the literature and to better embed the relevance of their findings in the field. This point is related to the above. Page 2, 4th paragraph "We began this project unsure of whether scalp EEG possesses the requisite fidelity to decode variability in mnemonic success, both during encoding and retrieval. Although prior intracranial EEG work suggested a positive outcome, the paucity of similar demonstrations using non-invasive recordings made us question whether scalp EEG data could support reliable trial-level classification." I don't think this statement is justified given the plethora of papers that demonstrate subsequent memory effects and memory recall effects using non-invasive electrophysiological methods. Here are a few examples of empirical papers (especially the first two seem very relevant here):

We agree with all of these points. We have significantly expanded our literature review to cite many of the papers noted by the reviewers. We have also removed or rewritten the offending text to address the reviewer's concerns.

Minor points

a. page 2, 3rd paragraph "Non-invasive EEG studies have found greater spectral tilt for successful as compared with unsuccessful memory encoding (Long et al., 2014; Long & Kuhl, 2019; Sederberg et al., 2006)." It took me a while to understand that greater tilt here means a counter clockwise rotation (i.e. low frequency power decreases and high frequency power increases). I always thought of it the other way

around, and other readers might be similarly confused. It would be good to specify what the authors mean with greater/smaller tilt right from the get-go to avoid this confusion.

We have clarified our discussion by specifying increased high-frequency activity and decreased alpha-band activity; we have added appropriate citations regarding these findings and also the mixed literature on theta modulations.

b. page 3, Participants. “... we excluded 10 subjects prior to analysis whose recall rates were between 15% and 85% for fewer than 10 sessions.” I did not understand this statement. Did you exclude subjects who were performing exceedingly poor or exceedingly good in at least 10 sessions? Please specify.

We have clarified this statement in the revised manuscript: "To ensure adequate representation of successful and unsuccessful mnemonic events we decided, prior to carrying out any analyses, to only include subjects with recall rates between 15% and 85% for 10 or more sessions."

c. page 4, Spectral Decomposition. “We log-transformed and normalized power across all events from the same task phase (encoding or retrieval) within each session.” Please specify exactly how power was normalized, ideally by providing a formula.

We z-scored the power estimates at each frequency x electrode pair across all events within each session. This was done separately for encoding events and retrieval events in each session. We have clarified this in the revised paper.

d. Please clarify that the classifier output is label prediction and not prediction accuracy.

We have added a clear reference to "classifier output probabilities" when we discuss the analyses using classifier outputs on the held-out test events.

e. Figure 4A and 4B the scatter plot is very hard to see. Perhaps reducing the transparency and scattering the individual datapoints in a way so they do not overlap would yield a better result. A distribution next to it would work as well. Currently the error bars and mean obstruct the view of the scatter plot.

Thank you for this suggestion. We have updated the AUC plots to show the distribution of the observed individual AUCs and the distribution of the mean baseline AUCs obtained from the permutation tests.

f. What is the advantage of using the L2-norm in the cost function? It is more susceptible to outliers compared to the L1-norm.

Our main rationale in using the L2-norm was to follow related earlier work applying logistic regression to decoding episodic encoding and retrieval successes in intracranial EEG (e.g., Ezzyat et al., 2017; Kragel et al., 2017; Ezzyat et al., 2018; Wiedemann et al., 2020). Per your suggestion, we re-examined our data by training the classifiers using an L1-norm (including all events; see figures below). We found that the L1-penalization led to slightly lower performance

but otherwise showed similar classifier output dynamics across event clustering subtypes and over list position/recall time. These L1-classifiers do appear to have excluded more input features used for prediction, leaving only LFA decrease for predicting encoding success and HFA increase predicting retrieval success. To avoid choosing between different analysis results, we have retained our original L2 classification results in our manuscript.

Encoding and retrieval classifiers trained with L1-norm on all events. Left, L1-classifier performance against L2-classifiers. Middle, feature activation maps associated with the L1 encoding and L1 retrieval classifiers. Right, L1 classifier outputs associated with different event subtypes.

g. I appreciate that the authors have published other papers where they investigate the behavioural data of this same dataset in more detail. But I would still like to see some numbers here regarding hits/misses, especially over time. This is also important when talking about the cost function and how class imbalance is counteracted by weighting the classes. The cost function includes a weight to balance out uneven trial numbers for example if there are more correctly remembered trials than prior list intrusions. How did you make sure that there are enough trials in each session to have a valid representation of the event? Is there an increase of PLI and ELI within a list and over lists? At least the former would be expected independent of attentional decrease. If there is one, how does such a bias effect the results?

We have revised our manuscript to include several new behavioral analyses including: 1) the probability of an encoding word being recalled over list positions (serial position curve; Figure 1A); 2) the probability density of recall responses over recall bins (Figure 1B), and the ratio of correct recalls and intrusions across recall bins (Figure 1B subpanel); 4) the probability of a subsequently recalled word being one of the subsequent clustering subtypes (Figure 6A); 5) the

probability of a correct recall response being one of the recall cluster or intrusion subtypes (Figure 6B);

We find that both prior- and extra-list intrusions increase slightly over the recall period. Although it is true that some single sessions that were held-out in training will not have any intrusion responses, we have verified that all training events (in each fold, including all sessions but the held-out session) had both positive and negative events. For participants who make very few intrusions, this does mean that the few instances of intrusions will be upweighted significantly to counter the class frequency, and could make the estimate noisy. However, validating on noisy estimates of intrusions would also hurt the AUC estimate in terms of large numbers of false positives. Our attempt to exclude participants based on recall rate was also an attempt to increase the probability that the training event contained a diversity of different event types.

h. How long were the “short breaks” every eight lists?

The breaks last about 3 minutes. We have added this information to the paper.

i. At one point the authors use a linear mixed effects model on the classifier performance to distinguish event types at different list positions. For the retrieval period the first and last 22.5 seconds were chosen as the early and late list positions. A close look at Figure 3A reveals that participants might not have a single response in the late period, whereas the majority of responses are given during the early period.

As shown in the newly-added Figure 1B, most recall responses occur relatively early in the recall phase. Evident from the classifier output figures (Figure 6D and 6F), outputs at the end of recall period, even when aggregated across participants, became quite noisy owing to the scarcity of recalls made in the last ~20 seconds. Nonetheless, we included these responses in our analyses to illustrate our findings across the entire recall period.

j. The abstract states that 98 participants performed the experiment. This includes the excluded participants. This should either be adapted in the abstract or made clear under Methods-> Participants that all subsequent analyses are based on 88 participants, which is still an impressive sample size.

We added a clarification statement at the end of the Participants subsection in Methods, after we discuss the exclusion criterion: "All reported analyses were based on the 88 participants who met this criteria."

k. In Figure 2A why does Subject 8 have a uniform difference power spectrum? Is this the color scaling?

The color scaling is centered at 0 (color white), with blue corresponding to decreases in power and red corresponding to increases in power. A few participants did show more universal increases or decreases across all frequencies when aggregated across ROIs.

l. Page 14, line 6. “In particular, classifier output was higher among BC ($M = 0.520$, $SE = 0.001$) and TC ($M = 0.517$, $SE = 0.001$) items compared to NC items ($M = 0.513$, $SE = 0.001$).“ <- significantly higher?

Although the values are very close in magnitude, the variance of the aggregated classifier output levels across participants in the two event subtypes was very small, and thus the mixed-effects model indicated that this difference was significant.

Reviewer #2 (Remarks to the Author):

The current study investigates EEG correlates of successful versus unsuccessful memory encoding and recall. The authors first characterize shifts in spectral power that occur during word encoding and preceding recall. They then use a pattern classification approach to test whether the observed correlates are sufficiently reliable to predict memory for individual trials on held out sessions. They find evidence for a spectral “tilt” whereby high frequency power is increased and low frequency power is decreased (with the exception of theta) for successful versus unsuccessful memory encoding and recall. Encoding and recall classification is generally very good, although different types of recall errors (prior vs extra list intrusions) cannot be reliably discriminated. Classifier evidence for successful memory also showed a primacy gradient and higher levels during encoding of words that would later be clustered by temporal and/or semantic context.

Overall, the paper is very well written and I find the results to be compelling (with some exceptions described below). Although there is no obvious theoretical advance here, the strengths of the paper lie in the “big data” approach. The study massively sampled both sessions and subjects resulting in remarkable statistical power that enables analyses that are not feasible with more traditional datasets. I think the approach moves the field forward, and although I would like to see more theoretically exciting questions addressed here, this is an important first step.

Major concerns:

My main concern is with the serial position and clustering analyses, which I find the most theoretically interesting. In particular, I’m concerned that the pattern of results could emerge based on bias in the training data. For example, if more training data is coming from earlier serial positions for recalled compared to not recalled, that could result in the primacy gradient observed simply because the classifier confounds subsequent memory with serial position. The same logic holds for retrieval and for the clustering analyses depending on the frequencies of BC, TC, etc. (also see comment below). To address this issue, the same permutation-based testing approach should be taken in this set of analyses as in the preceding analyses, which should address any potential bias.

Thank you for raising this excellent point, which mirrors comments of R1. As mentioned in the opening of this response letter, our new classification analyses based on event re-sampling prevents classifiers from relying on signals that vary across list positions in predicting mnemonic success. Analyzing the classifier output probabilities associated with the different event clustering subtypes using classifiers trained with re-sampled events thus provides a controlled test for the changing probability of different clustering types across serial positions or recall bins. The new results obtained using this method indicate that classifiers trained with event re-sampling exhibit greater sensitivity to subsequent clustering. However, as the reviewer anticipated, removing the

potential serial position confound attenuated the neural primacy effect evident in the classifier output (Compare revised Figures 6B and 6E).

Some description of the behavioral data is essential to include here. In particular, in order to interpret the neural data, we need to see frequency of PLI and ELI (a quick skim of the previous papers suggests this has yet to be described). Serial position curves mirroring Fig 6 and some characterization of temporal and semantic clustering is also needed.

Thank you for this suggestion. We have added a set of behavioral figures (Figures 1A and B, Figures 6A and B). Please refer to our response to Reviewer 1's point g for related comments.

Classifier evaluation: The within-subject permutation test is a strong test of classifier performance but the resulting null distributions should be shown, at least for the three example subjects. The binomial test for the group level data is less clear. Is each classifier (fold) treated independently? If so, nonindependence within subjects would violate key assumptions of the binomial distribution. I recommend at least one additional confirmatory analysis -- a simple group-level t-test comparing the actual and permutation-derived chance AUC values across subjects.

We have now added visualizations of the distribution of the baseline AUCs obtained from permuted data (Figures 4 and 5). We have also updated the binomial test to the t-test you suggested for the group-level tests, comparing the distribution of observed AUCs and the distribution of baseline AUCs (averaged across independent rounds permutations) for each classifier type.

Minor concerns/clarifications:

Classification of correct recall versus PLI and ELI is a subset of correct versus incorrect recall. The nonindependence of these analyses should be made more clear.

We have clarified this in the revised manuscript.

Is the increase in HFA really concurrent with the decrease in LFA? This claim should be supported by a time-frequency analysis.

The revised event-resampling analysis reported here indicates that LFA and HFA provide non-redundant information regarding mnemonic success. Below we illustrate the same univariate contrasts as in Figure 1D,E, but expanded to cover the 0-1600 ms encoding window and the 1000 ms window prior to recall vocalization. These time-frequency analyses indicate that whereas increased HFA and decreased LFA jointly appear during successful encoding when trained on all events, only LFA appears when controlling for the ratio of recalled and non-recalled events

across serial positions.

Time-frequency analysis of encoding and retrieval success for all events and the reviewer's suggested resampling to control for serial position effects. This analysis aggregates data across all eight ROIs.

How sensitive is classifier performance to the choice of regularization parameter? Why was the same parameter used for encoding and recall classifiers, and why was it based on classifying encoding states specifically? This choice might have influenced the observation that performance was somewhat more reliable for encoding compared to recall.

Our main rationale for using the encoding data to select the regularization parameter was that 1) each participant contributed roughly the same amount of data, which suits a leave-one-subject-out cross validation procedure for this hyperparameter search, 2) there were more encoding data and the EEG signals were likely less noisy, 3) our main goal was to optimize the regularization parameter given the nature of the EEG signal under the input dimension (number of frequency x number of channels), and we wanted to ensure more comparable results of the classifier output probabilities associated with the encoding and the retrieval classifiers. For these reasons, we transferred the value obtained from optimizing the prediction of encoding success to retrieval classifiers. The figure below shows the classifier performance across our 10 candidate regularization parameters during the hyperparameter search using the held-out 24th session data.

Classifier performance in predicting encoding success across data from the 24th session from all available participants, using a leave-one-subject-out cross-validation procedure, over candidate values of the regularization parameter. This was done separately for the two EEG recording systems.

The description of the encoding task is a little unclear. What were the instructions provided? If no response had to be made, was there anything done to ensure that the words were processed (e.g., eye tracking?)

We have revised our methods section to clarify that we did not instruct subjects to perform a specific task during item encoding. Rather, we simply asked subjects to study each item for a subsequent recall task. We have previously published free recall studies with and without encoding tasks (e.g., Polyn et al., 2009; Long et al., 2017). However, in the present experiment we sought to avoid having subjects perform a secondary task as such tasks may induce changes in the spectral components of the EEG.

In the intro, the authors suggest that reliable single-trial classification of memory states in scalp EEG data is quite limited. Although that seems largely true in episodic memory, single trial decoding is quite commonly demonstrated in the working memory literature. Thus, I would recommend qualifying that statement.

We have now highlighted that our contribution is on single trial decoding of episodic memory success in a free recall setting.

Describing EEG memory contrasts as biomarkers seems premature. Here the goal is simply to characterize the SME and recall effects whereas the term biomarker is typically used in a translational context to describe something with diagnostic relevance.

Thank you for pointing that out. We have changed the phrasing to "spectral markers".

Regarding the theta band results, the line, "supporting other recent studies linking lower theta band activity...with mnemonic success" is confusing. Does lower here refer to lower frequency, since the SME is actually associated with a power increase in theta?

Thank you for catching this confusing sentence. We have rewritten this sentence to clarify that we mean lower frequency theta band activity.

Reviewer #3 (Remarks to the Author):

In this manuscript, authors report decodability of scalp EEG signals for successful mnemonic events. Univariate analyses replicated the spectral tilt associated with successful encoding and retrieval, which has been shown in intracranial EEG studies from this group. Classifiers trained using power across frequency bands and electrodes as features reliably predicted subsequent recall from encoding data, and correct recall from signals prior to recall. Forward model constructed based on the classifiers revealed feature activation maps that resembled spectral tilts in univariate analysis. Encoding classifier outputs were stronger for early items in the word lists and for items that were recalled with temporal and semantic clustering. The findings that classifier outputs mirrored two very-well established, strong behavioral effects (primacy and clustering/item-context binding) provides good evidence that the classifiers were able to decode neural signals that underlie the exact mnemonic processes.

The study is theoretically well-grounded, and conducted with clear hypotheses and solid methods. The manuscript is also very well-written, and I only have a few suggestions and clarification questions for the authors.

1. Classifier outputs for recall activity were below chance level except for recalls that were made early in the retrieval phase (Figure 6B). Given the approaches taken, I'm having a hard time wrapping my head around why that would be the case. It would be worth to bring in the authors' take on this in the Discussion.

Thank you for raising this point. As illustrated in the new Figure 1B, subjects recalled many more words early in the retrieval period as compared with later in the retrieval period. As such, early points in the figure (the original Figure 6B, or the new Figure 6F) aggregated more events than later points, which means that at the level of single events, more correct recall responses would have a predicted probability above 0.5.

Relatedly, the original retrieval classifiers, which we trained on all events, can rely on signals that vary across the recall period to predict correct recalls. Because correct responses decrease while the intrusion responses increase over recall time, the retrieval classifier output will decline across the retrieval period. Classifiers trained with event re-sampling – which ensures matching ratio of positive and negative events from each recall bin – exhibit more consistent above-chance predictions for correct recall responses throughout the recall phase (see revised Figure 6B and 6C).

2. The following descriptions need to be clarified.

a. “we converted recordings to the average reference” This sounds like recordings were replaced with average reference, whereas authors probably meant that “recordings were off-line referenced to common average signals.”

We have clarified this language in the revised manuscript.

b. “we applied a fourth-order 4 Hz Butterworth notch filter at 60 Hz to attenuate electrical line noise.” It is unclear to me how a 4Hz filter can be applied at 60Hz – please clarify.

We filtered signals in a 4 Hz window centered at 60 Hz, attenuating signals between 58 Hz and 62 Hz. We have clarified this in the revised paper.

c. “We excluded peripheral electrodes (27 excluded for Geodesic caps, 18 excluded for BioSemi caps) close to the face and the neck in each system during all analyses.” Please provide specific criteria for peripheral electrodes (ideally with montages for the two systems), and justify using different numbers of electrodes for data from the two systems. I do not think this would have systematically influenced the results in any way, but as the number of electrodes impacts the number of classifier features and the consequent classifier performance, it would be important to spell out the details for transparency (it can go to appendix).

We now provide our analysis code in an open-source repository (<https://github.com/pennmem/EEG-memory-dynamics-public>), which includes a specific list of included electrodes, and the ROI-electrode mappings. We used different numbers of electrodes for the two systems mainly because the Geodesic caps, compared to the Biosemi caps, had 1) different number of electrodes on the cap, and 2) more electrodes that are closer to the face and the neck.

The reviewer correctly notes that the difference montages across the EGI and BioSemi systems resulted in different numbers of features used for classifications. We clarify in the revised manuscript (see page 6) that we optimized the regularization parameter separately for the two systems, and that results from this hyperparameter search suggested highly consistent patterns across the two systems. Please see our response to Reviewer 2's comment on the regularization parameter for related comments and figures.

d. “For memory search intervals in the free recall phases, we extracted the spectral patterns at 250 ms prior to speech onset, effectively covering a 500 ms memory search window before the observed onset of recall responses.” I’m confused about this. Was it the instantaneous spectral patterns (from precisely 250ms prior to recall) that were used here? If so, how does that ‘cover’ 500ms memory search window? It would fall in somewhere in the search window. Maybe the point being made here is that spectral patterns at 250ms prior to recall should be representative of the signals from the 500ms search window – but that is not clear.

We thank the reviewer for pointing out the need for clarification. As the reviewer notes, we intended to convey that the spectral estimate at 250 ms prior to vocalization onset is representative of signals from the 500 ms pre-vocalization period. We now clarify that our multi-taper method estimated the spectral power at 250 ms prior to vocalization using the EEG signals in the [-500 ms, 0 ms] window (with 0 ms corresponding to the onset of the recall response).

REVIEWERS' COMMENTS

Reviewer #1 (Remarks to the Author):

I want to thank the authors for addressing my comments. They have done a very good job to rule out the confound between memory success and list position. However, as I pointed out in my previous review, I feel that this paper is mostly replicating previous findings, which have shown similar findings also with non-invasive electrophysiology. I am therefore not sure about the conceptual advance that is offered by this study. It is certainly an interesting study that will be of great interest for memory researchers, especially those using electrophysiological methods, but I am not sure whether it is of great interest for the wider community, albeit this is a decision that ultimately the editor has to make.

Reviewer #4 (Remarks to the Author):

I agree that "The study is theoretically well-grounded, and conducted with clear hypotheses and solid methods."

The authors answered thoroughly all the questions of the original reviewer #3 and performed all the requested edits as requested by the original reviewer.

The authors should be applauded for adding a complete analysis of a resampled set of events, which equalized the items' list position, to reduce a potential confound of favorable positions. This allows the reader to dissociate the contribution of the serial position from the EEG correlates of mnemonic processes.

Suggestions:

The authors added the comparison to the re-sampled items list to every main figure, which I feel makes the manuscript more interesting when comparing the contribution of mnemonic and non-mnemonic processes to the classifier's success and is now a major part of the manuscript. However, the discussion wrapping up the comparisons between findings based on the two item-sets is very minimal (pg. 19) and I suggest adding clear statements about the differences between the all-events and the balanced-event position group.

Minor:

-- The colorbar in Figure 1 relates to colors in panels D-E but is placed in panel C, which is confusing.

-- The y-axis precision and limits are different between Figure 6 – panels E and F which makes it difficult to directly compare the results of the two classifiers.

Reviewer #1 (Remarks to the Author):

I want to thank the authors for addressing my comments. They have done a very good job to rule out the confound between memory success and list position. However, as I pointed out in my previous review, I feel that this paper is mostly replicating previous findings, which have shown similar findings also with non-invasive electrophysiology. I am therefore not sure about the conceptual advance that is offered by this study. It is certainly an interesting study that will be of great interest for memory researchers, especially those using electrophysiological methods, but I am not sure whether it is of great interest for the wider community, albeit this is a decision that ultimately the editor has to make.

Reviewer #4 (Remarks to the Author):

I agree that “The study is theoretically well-grounded, and conducted with clear hypotheses and solid methods.”. The authors answered thoroughly all the questions of the original reviewer #3 and performed all the requested edits as requested by the original reviewer.

The authors should be applauded for adding a complete analysis of a resampled set of events, which equalized the items’ list position, to reduce a potential confound of favorable positions. This allows the reader to dissociate the contribution of the serial position from the EEG correlates of mnemonic processes.

Suggestions:

The authors added the comparison to the re-sampled items list to every main figure, which I feel makes the manuscript more interesting when comparing the contribution of mnemonic and non-mnemonic processes to the classifier’s success and is now a major part of the manuscript. However, the discussion wrapping up the comparisons between findings based on the two item-sets is very minimal (pg. 19) and I suggest adding clear statements about the differences between the all-events and the balanced-event position group.

Response: We appreciate this comment and have expanded the general discussion to discuss these differences (see Page X, lines Y-Z of the revised manuscript).

Minor:

- The colorbar in Figure 1 relates to colors in panels D-E but is placed in panel C, which is confusing.
- The y-axis precision and limits are different between Figure 6 – panels E and F which makes it difficult to directly compare the results of the two classifiers.

Response: We have made each of the suggested revisions